# GAAVI: Global Asymptotic Anytime Valid Inference
# for the Conditional Mean Function

**Brian M Cho** [* 1]  **Raaz Dwivedi** [1]  **Nathan Kallus** [1 2]

## Abstract

Inference on the conditional mean function (CMF) is central to tasks from adaptive experimentation to optimal treatment assignment and algorithmic fairness auditing. In this work, we provide a novel asymptotic anytime-valid test for a CMF global null (e.g., that all conditional means are zero) and contrasts between CMFs, enabling experimenters to make high confidence decisions at *any* time during the experiment beyond a minimum sample size. We provide mild conditions under which our tests achieve (i) asymptotic type-I error guarantees, (i) power one, and, unlike past tests, (iii) optimal sample complexity relative to a Gaussian location testing. By inverting our tests, we show how to construct function-valued asymptotic confidence sequences for the CMF and contrasts thereof. Experiments on both synthetic and real-world data show our method is well-powered across various distributions while preserving the nominal error rate under continuous monitoring.

## 1. Introduction

Inference on the conditional mean function (CMF) is central to decision making in a wide range of modern applications, including public health (Segal et al., 2023; Bellavia & Murphy, 2024), algorithmic monitoring (Chugg et al., 2025a), and online platforms (Johari et al., 2019). In algorithmic audit applications, inference on conditional regression/classification outputs across sensitive attributes enables tests for fairness metrics such as statistical parity (Dwork et al., 2011). In randomized controlled trials, the conditional average treatment effect (CATE) plays a central role in assessing treatment effect heterogeneity (Künzel et al., 2019) and learning optimal policies (Athey & Wager, 2020).

[1]Department of ORIE, Cornell Tech, New York, NY, USA [2]Netflix, New York, NY, USA. Correspondence to: Brian M Cho <bmc233@cornell.edu>.

*Proceedings of the 43rd International Conference on Machine Learning*, Seoul, South Korea. PMLR 306, 2026. Copyright 2026 by the author(s).

As online data collection and continuous monitoring become increasingly common (Johari et al., 2019), there is a growing need for sequential tests that provide online, flexible inference on CMFs for timely, high-confidence decision-making. In many settings, the ability to conduct inference on the CMF *during* the data collection process is particularly valuable (Chugg et al., 2025a; Adam et al., 2023).

For example, fairness constraints may require practitioners to halt deployment if a model's output differs across sensitive attributes (Chugg et al., 2025a). With sequential tests, practitioners can identify such disparities with high confidence as observations accumulate and halt deployment accordingly. Similarly, in clinical trials, sequential inference on the CATE can be critical for patient safety. For drugs such as warfarin, a positive average treatment effect may mask adverse effects for subgroups, such as elderly patients (Shendre et al., 2018). The ability to test the CATE function sequentially allows for early stopping, mitigating potential harm to negatively affected groups with high confidence.

Despite its practical importance, existing works on CMF testing largely focus on (i) offline settings (Liu et al., 2023; Arias-Castro et al., 2011), where inference is conducted at a fixed sample size, and/or (ii) finite context sets, where the CMF is restricted to being a vector/matrix (Howard et al., 2021; Cho et al., 2024a). In contrast, relatively little attention has been paid to sequential inference on the CMF in settings for more general, potentially finite conditioning sets, limiting the applicability of current methods. Without discretizations of the context/covariate space (Ghosh et al., 2022) or parametric model projections (Lindon et al., 2025), existing methods fail to provide high-confidence conclusions for testing the CMF during data collection when covariates/contexts are continuous or high-dimensional.

**Contributions.** We construct nonparametric asymptotic anytime-valid tests for the CMF/CATE under a general covariates/context space. Our tests are designed for the *global null*, which assesses whether the CMF/CATE is equal to a prespecified conditional mean function of interest. Building upon the asymptotic anytime validity framework, our tests use a novel weighted martingale that leverages flexible regression methods to maximize power against the specified global null. Using our tests, we also construct function-

valued asymptotic confidence sequences for the CMF/CATE. Importantly, our tests are provably robust, achieving

(i) **asymptotic anytime valid error control** even if *no* nuisances/regressors used in our test converge,

(ii) **power one** under a mild positive covariance condition,

(iii) and asymptotically **optimal sample complexities** when nuisances converge to the desired quantities.

For global null testing, under suitable regularity, our results show nonparametric sequential testing with asymptotic error guarantees is no harder than non-asymptotic sequential testing for conditional Gaussian models with known variance. Our experiments show our method is well-powered across a variety of data generating processes relative to existing methods, while preserving error guarantees.

**Outline.** The paper is organized as follows. In the remainder of this section, we briefly cover related works. In Section 2, we formalize our problem for CMF and CATE testing, focusing on the global null hypothesis. In Section 3, we introduce our approach for testing the global null and demonstrate how to leverage these tests for confidence sequences on the CMF/CATE. In Section 4, we provide conditions under which our tests maintain error guarantees, obtain power one, and achieve optimal sample complexities. In Section 5, we test our method against related approaches, empirically demonstrating robust detection power while maintaining error rates. We present our conclusions in Section 6.

## 1.1. Related Works

We provide a brief overview of related work, focusing on (i) asymptotic anytime validity and (ii) global null testing.

**Asymptotic Anytime Valid Inference.** Recent years have seen the rise of anytime valid (AV) testing methods (Ramdas et al., 2023; Ramdas & Wang, 2025). Our work most closely builds upon the line of work called asymptotic anytime-valid inference (Bibaut et al., 2024; Waudby-Smith et al., 2024; Kilian et al., 2025; Cho & Kallus, 2025), which leverages strong invariance principles to obtain approximate time-uniform error control. Instead exact time-uniform control, asymptotic AV inference provide approximate error control over all times beyond a sufficiently large minimum sample size (i.e. burn-in time). While our work leverages similar tools to ensure asymptotic anytime validity, the focus of our work differs from the goals of existing work. Many asymptotic AV methods (Bibaut et al., 2024; Waudby-Smith et al., 2024; Kilian et al., 2025) focus on point null tests for a univariate mean parameter, rather than for the entire conditional mean functions, and use their tests to form asymptotic confidence sequences for the mean of interest. The closest work to our testing procedure is Cho & Kallus (2025),

where the authors construct asymptotic AV tests identifying the highest mean treatment as opposed to inference on a scalar parameter. In contrast, our tests provide asymptotic AV error control for the global null and related hypotheses.

**Global Null Testing.** Global null testing has a rich history in statistics, focusing on assessing whether conditional means are identically zero (or equal to a known function) across the entire context space (Fisher, 1932; Ingster, 1989; Ingster & Suslina, 2003). Classical approaches for testing the global null in linear models include $F$-tests/ANOVA (Fisher, 1932), higher criticism methods (Donoho & Jin, 2004), and max tests based on multiple testing corrections (Arias-Castro et al., 2011). Beyond linear models, works such as Ingster & Sapatinas (2009) provide tests under more flexible, nonlinear regression setups. Our work is most connected to tests which combine information into a single test statistic (Fisher, 1932), rather than multiple testing corrections (Donoho & Jin, 2004). However, existing methods typically assume parametric assumptions (e.g. Gaussian error), require sample sizes to be fixed in advance, and do not provide guarantees under continuous monitoring. In contrast, our tests are fully nonparametric, do not require fixed sample sizes, and provide approximate guarantees under continuous monitoring for sufficiently large sample sizes.

**Anytime Valid Testing for the Global Null.** While sequential global null tests with AV guarantees exist, they fail to accommodate fully nonparametric inference under general context spaces or differ substantially in aim. Works such as Chugg et al. (2025b); Howard et al. (2021) enable nonparametric AV inference for finite context spaces, where martingale methods can be directly applied. While Lindon et al. (2025) provides AV inference for continuous, multivariate contexts, these methods focus on testing linear regression coefficients, reducing the problem to a finite dimensional setting and only providing inference on the best linear projection of the CMF/CATE. Lastly, while the work of Duan et al. (2021) also study sequential inference on the global null, they provide non-asymptotic AV inference under the generic setting where hypotheses arrive sequentially. In contrast to existing work, our work specifically focuses on the CMF/CATE, provides asymptotic AV error guarantees, allows for continuous/high dimensional contexts without parametric projections, and establishes conditions for achieving asymptotically optimal sample complexities.

## 2. Problem Setup and Preliminaries

In our work, we assume that observations $(O_i)_{i \in \mathbb{N}}$ arrive sequentially, where $O_i$ are i.i.d. observations from an unknown distribution $P$. To accommodate our examples, we provide two distinct data generating processes (DGPs) corresponding to (i) generic CMF testing, denoted as DGP1 and (ii) CMF contrast testing in online experiments with binary

treatments, which we denote as CATE testing (DGP2).

**DGP1.** We assume $O_i = (X_i, Y_i)$ and $P = P_X \times P_{Y|X}$, where $X_i \in \mathcal{X}$ denotes the context (i.e. the conditioning variable) and $Y_i \in \mathbb{R}$ denotes the outcome of interest. Our object of interest is the CMF $\tau(x) = \mathbb{E}_{P_{Y|X}}[Y|X = x]$.

**DGP2.** We assume $O_i = (X_i, A_i, Y_i)$ and $P = P_X \times P_{A|X} \times P_{Y|A,X}$. For each observation, the context $X_i \in \mathcal{X}$ is generated i.i.d. from an unknown distribution $P_X$. After observing the context $X_i$, the experimenter assigns a binary treatment $A_i \in \{0, 1\}$, where $A_i$ is sampled according to a known policy $\pi(X_i, a) = P(A_i = a|X = X_i)$, corresponding to a randomized experiment. The outcome of interest $Y_i \in \mathbb{R}$ is then generated according to an unknown conditional distribution $P_{Y|X,A}$. We note that knowledge of the sampling policy $\pi$ is a standard assumption in online experiments (Cho et al., 2025; Adam et al., 2023), and for simplicity, we assume that $\pi$ is fixed over time. We denote the CATE function, our object of interest, as $\tau(x) = \mu(x, 1) - \mu(x, 0)$, where $\mu(x, a) = \mathbb{E}_{P_{Y|A,X}}[Y|A = a, X = x]$.[1]

For both DGPs, we define the conditional variance function

$$\sigma^2(z) = \mathbb{E}_P\left[(Y - \mathbb{E}_P[Y|Z = z])^2 | Z = z\right], \quad (1)$$

where $Z := X$ and $Z := (X, A)$ for DGP1 and DGP2 respectively. As shorthand, we denote all observations up to time $t$ as $H_t = (O_i)_{i \in [t]}$, where $[t] = \{1, \ldots t\}$. We use $\mathcal{F}_t = \sigma(H_t)$ to denote the canonical filtration, with $\mathcal{F}_0$ as the trivial, empty sigma field. We denote the set of functions $L_\infty(B)$ as the set of all bounded functions $f : \mathcal{X} \to [-B, B]$. We make the following assumptions on both DGPs regarding variances and outcomes.

**Assumption 2.1** (Positive Conditional Variances). There exists $b > 0$ such that variances $\sigma^2(z) \geq b$ for all $z \in \mathcal{Z}$.

Assumption 2.1 ensures that our sample complexity bounds scale with our type I error tolerance and avoids degeneracy issues. Note that Assumption 2.1 also implies that marginal outcome variances are nonzero. We also assume that the outcome $Y$ is bounded, formalized in Assumption 2.2.

**Assumption 2.2** (Bounded Outcomes). There exists a (potentially unknown) constant $B$ such that $|Y| \leq B$.

In most common applications, Assumption 2.2 is likely to hold, even if the maximum magnitude of the outcome variable is unknown. In contrast with nonasymptotic AV methods (Chugg et al., 2025b), which require outcome bounds $B$ (or moment bounds) to be known in advance, our tests do

---

[1]To enable valid causal interpretations under the Neyman-Rubin potential outcomes model, we require additional conditions such as consistency and SUTVA. Because our work is primarily focused on global null tests, we refer readers to Appendix B for more detailed discussion.

not require this constant $B$ to be known or estimated. For DGP2, we also assume that the following holds.

**Assumption 2.3** (Strict Positivity). There exists $\kappa < \infty$ such that $\pi(a, x) \geq 1/\kappa$ for all $a \in \{0, 1\}$ and $x \in \mathcal{X}$.

Similar to the outcome bound $B$, the inverse propensity score upper bound $\kappa$ does not need to be known apriori to data collection. Other than the assumptions provided above, we make *no further assumptions* on the DGP, allowing for highly nonparametric, complex outcome distributions.

### 2.1. Problem Statement

The goal of our work is to construct asymptotic anytime-valid tests regarding CMF/CATE function $\tau$. To begin, we first define asymptotic anytime validity in Definition 2.4.

**Definition 2.4** (Asymptotic Anytime Validity). Let $\mathcal{H}_0$ be a collection of distributions $P$, and let $\xi : H_\infty \times (0, 1) \to \{0, 1\}$ be a mapping from the observed data stream and nominal error level to a binary output. Then, the function $\xi : H_\infty \times (0, 1) \to \{0, 1\}$ is an $\alpha$-level asymptotic anytime-valid test for the null hypothesis $\mathcal{H}_0$ if $\forall P \in \mathcal{H}_0$,

$$\limsup_{t_0 \to \infty} P\left(\exists t \geq t_0 : \xi(H_t, \alpha) = 1\right) \leq \alpha. \quad (2)$$

Our definition ensures that error rate is approximately controlled uniformly over times $t \geq t_0$, where $t_0$ denotes a burn-in time that specifies the minimum sample size before rejection. Tests satisfying Definition 2.4 enable real-time decision making: rejecting the null at any time $t \geq t_0$ implies that, for sufficiently large $t_0$, the null is false with probability at least $1 - \alpha$, allowing experimenters to draw high-confidence conclusions throughout the experiment.

In this work, we focus on constructing tests that satisfy Definition 2.4 for the global null hypothesis $\mathcal{H}(f) : \tau(x) = f(x)$, which tests equivalence of $\tau$ to a known function $f$. Below, we provide practical examples corresponding to the global null across various applications.

**Examples for DGP1.** Let $X \in [0, 1]$ be predictive probabilities from a pretrained model $\nu$ for a binary outcome $Y \in \{0, 1\}$. The null hypothesis $\mathcal{H}(x) : \tau(x) = x$ corresponds to the hypothesis that model $\nu$ is calibrated (Dawid, 1982; Guo et al., 2017), i.e. the predicted probabilities from $\nu$ match the expected frequency (w.r.t $P$). Rejection of the null implies that the model $\nu$ is not $P$-calibrated.

**Examples for DGP2.** When $\tau$ is the CATE, $\mathcal{H}(0) : \tau(x) = 0$ corresponds to the null that no $X$-defined subgroups have a treatment effect, matching the classical global null hypothesis setup (Arias-Castro et al., 2011). Rejection of the null implies there exists at least one subgroup with a treatment effect. In the setting of algorithmic fairness, where $Y \in \{0, 1\}$ is an algorithmic recommendation, $A \in \{0, 1\}$

denotes a sensitive attribute (with known conditional population prevalence), and $X$ denotes factors of interest, $\mathcal{H}(0)$ corresponds to the fairness notion of *conditional statistical parity* (Dwork et al., 2011; Kamiran et al., 2013).

## 3. Testing the Conditional Mean Function

For both DGPs, we build sequential tests for the null $\mathcal{H}(f)$ by constructing $(\mathcal{F}_t)_{t\in\mathbb{N}}$-adapted martingales with zero mean increments under the null. We do so with the terms

$$\phi_i^{CMF} = Y_i, \quad \phi_i^{CATE} = \phi_{i,1} - \phi_{i,0}, \tag{3}$$

where $\phi_{i,a} = g_i(X_i, a) + \frac{\mathbf{1}[A_i = a](Y_i - g_i(X_i, a))}{\pi(X_i, a)}$ and $g_i$ is an $\mathcal{F}_{i-1}$-measurable estimate of conditional means $\mu(x, a)$ in DGP2. The terms $\phi_i^{CMF}$ and $\phi_i^{CATE}$ act as conditionally unbiased terms for point evaluations of the CMF/CATE respectively, i.e. $\mathbb{E}_P[\phi_i | X_i = x, \mathcal{F}_{i-1}] = \tau(x)$ for their respective functions of interest. To ease exposition, we drop the superscripts on $\phi_i$ to refer to either DGP1 and DGP2, and provide a unified exposition for constructing our test.

Leveraging the conditional unbiased property, we construct the process $\psi_t(f)$ by taking weighted sums of the difference between $\phi_i$ and the null CMF/CATE function $f$,

$$\psi_t(f) = \sum_{i=1}^{t} w_i(X_i, f)\, (\phi_i - f(X_i)), \tag{4}$$

where $w_i(x, f)$ is an $\mathcal{F}_{i-1}$-measurable weight function. As shorthand, we denote $\bar{\psi}_t(f) = \psi_t(f)/t$ as the time-normalized process. Note that by the conditional unbiased property, for any distribution $P \in \mathcal{H}(f)$, each term $w_i(X_i, f)\,(\phi_i - f(X_i))$ in our running sum $\psi_t$ has zero conditional mean for any choice of weights $w_i$, ensuring that the process $\psi_t$ is a martingale w.r.t. $(\mathcal{F}_t)_{t\in\mathbb{N}}$ and $P \in \mathcal{H}(f)$.

We leverage the fact that process $\psi_t$ has zero conditional mean for any $P \in \mathcal{H}(f)$ to reject the null. To obtain the desired error control in Definition 2.4, we construct asymptotic AV bounds $L_t(f)$ for the average conditional mean of the increments of process $\psi_t$. Similar to other asymptotic AV approaches (Bibaut et al., 2024; Waudby-Smith et al., 2024; Cho & Kallus, 2025), we leverage bounds based on the Gaussian mixture martingale, which includes (i) parameter $\rho > 0$, corresponding to a prespecified time $t^*$ where $L_t(f)$ is tightest[2], and (ii) variance term $\hat{V}_t(f)$, which aims to estimate the average conditional variance of the martingale difference terms $w_i(X_i, f)\,(\phi_i - f(X_i))$ in our process $\psi_t$:

$$L_t(f, \alpha, \rho) = \bar{\psi}_t - \ell_{t,\alpha\rho}\left(\hat{V}_t(f)\right), \tag{5}$$

$$\ell_{t,\alpha,\rho}(x) = \sqrt{\frac{2(tx\rho^2 + 1)}{t^2\rho^2} \log\left(1 + \frac{\sqrt{tx\rho^2 + 1}}{2\alpha}\right)}. \tag{6}$$

---

[2]We provide standard choices of $\rho > 0$ in Appendix B.3.

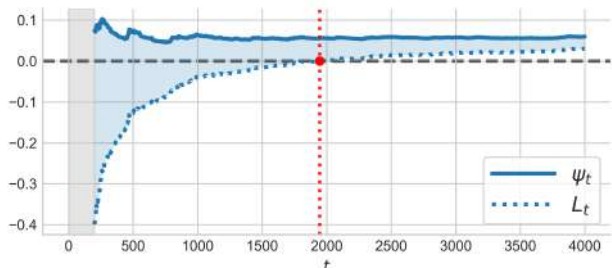

*Figure 1.* Visualization of our method for one instance with $t_0 = 200$ (grey region). The red dotted line indicates time of rejection.

Beyond sufficiently large burn-in times $t_0$, the lower bound sequence $(L_t(f, \alpha, \rho))_{t\in\mathbb{N}}$ serves as a time uniform $1 - \alpha$ lower bound for the average conditional means of terms $w_i(X_i, f)\,(\phi_i - f(X_i))$. If $L_t$ crosses above zero at any time $t \geq t_0$, the asymptotic anytime valid guarantees of $L_t$ ensure the cumulative average drift of $\psi_t(f)$ is positive and nonzero with probability at least $1 - \alpha$.

Because the drift of $\psi_t(f)$ is strictly zero when $P \in \mathcal{H}(f)$, the probability that the lower bound $L_t(f, \alpha, \rho)$ rises above zero beyond sufficiently large $t_0$ for any $P \in \mathcal{H}(f)$ is less than $\alpha$. Thus, our sequential test $\xi_t(f, \alpha, \rho, t_0)$ is simply

$$\xi_t(f, \alpha, \rho, t_0) = \mathbf{1}\left[\max_{t_0 \leq i \leq t} L_i(f, \rho, \alpha) > 0\right], \tag{7}$$

i.e. reject the null $\mathcal{H}(f) : \tau(x) = f(x)$ as soon as the lower bound process $L_t(f, \rho, \alpha)$ crosses above zero for $t \geq t_0$. In Figure 1, we visualize our testing procedure for clarity.

### 3.1. Predictable Weights for GAAVI

Our choice to track lower bounds, rather than a two-sided confidence sequence for the average conditional drift of $\psi_t(f)$, follows from our predictable weight function $w_i$. The predictable weights $w_i$ take the form

$$w_i(X_i, f) = \text{sgn}(\tilde{w}_i(X_i, f)) \max\{\epsilon_i, |\tilde{w}_i(X_i, f)|\}, \tag{8}$$

$$\tilde{w}_i(X_i, f) = \frac{\hat{\tau}_i(X_i) - f(X_i)}{\hat{v}_i(X_i)}, \tag{9}$$

where $\hat{\tau}_i(x)$ and $\hat{v}_i(x)$ denote $\mathcal{F}_{i-1}$-measurable estimates of our function of interest $\tau$ and the $X$-conditional variance of the terms $\phi_i$ respectively. The weight function $\tilde{w}_i$ corresponds to the difference between $\hat{\tau}_i(x)$, an estimate of $\tau(x)$ with data $H_{i-1}$, and $f(x)$, the value of $\tau(x)$ under our null, divided by an estimate of conditional variances $v_i(x) = \mathbb{E}_P[(\phi_i - \tau(x))^2 | X = x, \mathcal{F}_{i-1}]$. The weight function $w_i$ then thresholds the magnitude of $\tilde{w}_i$, ensuring that the weight magnitudes remain larger than $\epsilon_i$.

Our weights directly correspond to our choice to *only* track the lower bound on the average drift of $\psi_t(f)$. Consider an

oracle version of our weight $\hat{w}_i$ with $\hat{\tau}_i(X_i) = \tau(X_i)$ in the numerator, $\hat{v}_i(X_i) = v_i(X_i)$ in the denominator, and $\epsilon_i = 0$. Then, the conditional drift $\mathbb{E}\left[\psi_i(f) - \psi_{i-1}(f) | \mathcal{F}_{i-1}\right]$ with our oracle weights has nonnegative drift at time $i$, i.e.

$$\mathbb{E}_{P_X}\left[\frac{(\tau(X) - f(X))^2}{v_i(X)} \Big| \mathcal{F}_{i-1}\right] \geq 0. \quad (10)$$

If there exists some $\hat{\mathcal{X}} \subseteq \mathcal{X}$ with positive measure (w.r.t. $P_X$) where $\tau(x)$ and $f(x)$ differ, then the conditional drift is strictly positive. Thus, our weight scheme aims to add nonnegative drift to $\psi_t(f)$ when the null is false, corresponding to our choice to construct lower bounds for our test $\xi_t$.

*Remark* 3.1 (On Thresholds and Variance Weighing). While the numerator of $w_i(X_i, f)$ justifies our choice to track lower bounds, our weights also involve (i) an inverse estimated variance term $\hat{v}_i^{-1}$ and (ii) magnitude thresholds. The former corresponds to our optimal sample complexity results in Theorem 4, Section 4. The latter corresponds to a technical condition that ensures cumulative variances diverge, enabling Gaussian asymptotic approximation. We provide guidance on the choice of our weight magnitude bounds $\epsilon_i$ in Theorem 4.2, Section 4.

### 3.2. Variance Estimation for Sequential Testing

In the design of our test, we leverage two distinct variance estimates: (i) the term $\hat{V}_i(f)$, which estimates the average conditional variance of the martingale difference terms $w_i(X_i, f)(\phi_i - f(X_i))$, and (ii) the $\mathcal{F}_{i-1}$-measurable variance function $\hat{v}_i(X)$ used for our predictable weights $w_i$.

To estimate these quantities, we use empirical counterparts that center terms $\phi_i$ using an *estimated* function $\hat{\tau}_i$, rather than centering using the null $f$. For the average conditional variance terms $\hat{V}_t(f)$, we estimate this quantity with

$$\hat{V}_t(f) = \frac{1}{t}\sum_{i=1}^{t} w_i^2(X_i, f) r_i^2, \quad r_i = (\phi_i - \hat{\tau}_i(X_i)). \quad (11)$$

Our running estimate $\hat{V}_t(f)$ is simply the squared sum of terms $\phi_i$ centered with an $\mathcal{F}_{i-1}$-measurable estimate $\hat{\tau}_i$ of function $\tau$. Similarly, to obtain our $\mathcal{F}_{i-1}$-measurable estimator $\hat{v}_i$, we construct the function $\tilde{v}_i(x)$ by regressing the squared estimated residual terms $\{r_j^2\}_{j<i}$ against $\{X_j\}_{j<i}$. To avoid degeneracy issues when constructing $w_i$ in Equation (8), we clip our the function $\tilde{v}_i$ to obtain

$$\hat{v}_i(X) = \max\{\tilde{v}_i(X), l\}, \quad (12)$$

where $l > 0$ is a small constant that bounds estimated conditional variances $\hat{v}_i$ away from zero.

*Remark* 3.2 (Comparisons with Sample Variance). Standard methods that leverage asymptotic AV lower bounds (Bibaut et al., 2024; Cho & Kallus, 2025) use the sample variance

*Table 1.* Parameters/Nuisances for $\xi_t$ and their roles.

| Variable | Role |
|---|---|
| $\alpha \in (0, 1)$ | error tolerance |
| $\rho \in \mathbb{R}_{++}$ | tightening parameter for $L_t$ at $t^*$ |
| $t_0 \in \mathbb{N}$ | minimum sample size / burn-in time |
| $g_t : (\mathcal{X}, \{0, 1\}) \to \mathbb{R}$ | $\mathcal{F}_{t-1}$-measurable regression function for estimating $\mu(x, a)$ in DGP2 |
| $\hat{\tau}_t : \mathcal{X} \to \mathbb{R}$ | $\mathcal{F}_{t-1}$-measurable regression function for estimating CMF/CATE $\tau$ |
| $\hat{v}_t : \mathcal{X} \to \mathbb{R}$ | $\mathcal{F}_{t-1}$-measurable regression function for estimating conditional variance $\mathbb{E}[(\phi - \tau(x))^2 | X = x]$ |
| $\epsilon_t \in \mathbb{R}_{++}$ | magnitude bound for sequential weights $w_t(X_t, f)$ in Equation (8) |
| $l \in \mathbb{R}_{++}$ | lower bound on variance function $\hat{v}_t$ |

of the martingale difference terms to construct $\hat{V}_t(f)$. In our work, the choice of $\hat{V}_t(f)$ differs due to the fact that we wish to estimate the variance *as if the null were true*, even if $P \notin \mathcal{H}(f)$. While the sample variance converges to variance under the null when $P \in \mathcal{H}(f)$, the limit of sample variance is larger than the variance under the null when $P \notin \mathcal{H}(f)$ by the factor $\mathbb{E}[(\tau(x) - f(x))^2]$, the average squared difference between $\tau$ and $f$. By centering the $\phi_i$ terms with running estimates of $\tau$, $\hat{V}_t(f)$ converges to the desired variance under the null if $\hat{\tau}_i$ converges to $\tau$, and provides a conservative (i.e. larger) estimate otherwise.

To enumerate the necessary parameters and conditional regression functions (i.e. nuisances) used for our test $\xi_t$, we provide Table 1, which specifies parameter values, nuisance mappings, and their role in our testing procedure $\xi_t$.

### 3.3. Generalization to Confidence Sequences

Using the tests for the null $\mathcal{H}(f)$, we construct asymptotic confidence sequences for the CMF/CATE function. Our sequences $C_t(\rho, \alpha, t_0)$ are simply inversions of our tests $\xi_t$,

$$C_t(\rho, \alpha, t_0) = \{f \in L_\infty(B) : \xi_t(f, \alpha, \rho, t_0) = 0\}, \quad (13)$$

where $C_t(\rho, \alpha, t_0)$ is the set of bounded functions $f \in L_\infty(B)$ that have not yet rejected by time $t$ by $\xi_t$.

When $\mathcal{X}$ is finite and bounds $B$ are known, $C_t(\rho, \alpha, t_0)$ can be constructed by explicitly sweeping over the hypothesis space $L_\infty(B)$. However, when $\mathcal{X}$ is continuous or high-dimensional, explicitly tracking confidence sequences is generally infeasible or computationally prohibitive. In such settings, rather than forming confidence sequences, our tests most naturally apply to pre-specified candidate CMF/CATEs or tracking whether $C_t$ intersects with nulls of interest, such as $\tau(x) = c$ for some constant $c$. The latter example can be efficiently implemented with univariate function minimizers readily available in standard software, such as Brent (1973).

*Remark* 3.3 (Choice of Regression Functions). Our testing procedure involves multiple sequential regressors/nuisances.

For both DGP1 and DGP2, we require leverage $\mathcal{F}_{t-1}$-measurable regressors $\hat{\tau}_t$ and $\hat{v}_t$ that aim to estimate conditional means $\tau(X_i) = \mathbb{E}[\phi_i|X = X_i]$ and variances $v(X_i) = \mathbb{E}[(\phi_i - \tau(X_i))^2 |X = X_i, \mathcal{F}_{i-1}]$ respectively. For DGP2, we require $\mathcal{F}_{t-1}$-measurable regression functions $\hat{g}_t$ that aims to estimate conditional means $\mu(X_i, a) = \mathbb{E}[Y|X = X_i, A = a]$ in order to construct $\phi_i$. Our test accommodates a broad class of sequential regression estimators for estimating $\hat{\tau}_t$, $\hat{v}_t$, and $g_t$, including flexible nonparametric methods such as $k$-NN (Yang & Zhu, 2002), kernel methods (Qian & Yang, 2016), random forests (Wager & Athey, 2018), and neural networks (Schmidt-Hieber, 2020). Simpler parametric approaches such as linear/logistic regression may also be used. In Section 4, we provide sufficient conditions on our regression functions for error control, power one, and optimal sample complexity.

# 4. Theoretical Guarantees

We provide sufficient conditions under which our tests $\xi_t$ achieve (i) asymptotic AV error control (as in Definition 2.4), (ii) power one, and (iii) optimal sample complexities relative to a Gaussian shift model. We first introduce mild regularity condition regarding the boundedness of our sequential regressors sufficient for type I error control.

**Assumption 4.1** (Boundedness of Nuisances). For all $t \in \mathbb{N}$, $x \in \mathcal{X}$, and $a \in \{0, 1\}$, there exists some constant $B < \infty$ such that $|\hat{\tau}_t(x)| \le B$, $|g_t(x, a)| \le B$, and $\hat{v}_t(x) \le B^2$.

Assumption 4.1 naturally follows from our bounded outcome assumption in Assumption 2.2. Similar to the role of the bounds in Assumption 2.2, we note that this constant $B$ does not need to be known or estimated, and only plays a role in our theoretical guarantees. Under the boundedness conditions of Assumption 4.1, our tests $\xi_t$ provide asymptotic AV error guarantees for their respective nulls.

**Theorem 4.2** (Asymptotic Anytime Validity of $\xi_t$). *Let Assumptions 2.1- 2.3 and 4.1 hold. Then, for fixed error tolerance $\alpha \in (0, 1)$, $\rho > 0$, conditional variance threshold $l > 0$ (as defined in Equation (12)), and weight bounds $\epsilon_t = \Omega(t^{-\gamma})$ for $\gamma \in [0, 1/4]$ (as defined in Equation (8)),*

$$\lim_{t_0 \to \infty} P\left(\exists t \ge t_0 : \xi_t(f, \alpha, \rho, t_0) = 1\right) \le \alpha \quad (14)$$

*for all distributions $P \in \mathcal{H}(f)$ and $f \in L_\infty(B)$. Likewise, under the same conditions on parameters $\alpha$, $\rho$, $l$, and $\epsilon_t$,*

$$\lim_{t_0 \to \infty} P\left(\exists t \ge t_0 : \tau \notin C_t(\rho, \alpha, t_0)\right) \le \alpha, \quad (15)$$

*where $C_t(\rho, \alpha, t_0)$ is as defined in Equation (13).*

Theorem 4.2 establishes that our tests $\xi_t$ satisfy the asymptotic AV error guarantee in Definition 2.4 under mild boundedness assumptions (Assumptions 2.1–2.3 and 4.1) and the condition that the weight bounds $\epsilon_t$ decay at rate $\Omega(t^{-1/4})$. In particular, the constraints on variance bound sequence $(\epsilon_t)_{t \in \mathbb{N}}$ ensure our process $\psi_t$ is (i) well approximated by a scaled Wiener process (Strassen, 1964), allowing the Gaussian mixture martingale $L_t$ to provide asymptotic guarantees, and (ii) polynomial rates of estimation for the running variance $\hat{V}_t$, which ensures our bounds protect the error rate. Notably, the results of Theorem 4.2 do not require convergence of sequential regressors $\hat{\tau}_t$, $\hat{g}_t$, or $\hat{v}_t$, demonstrating that $\xi_t$ protects type I error under very general conditions.

## 4.1. Tests of Power One

While Theorem 4.2 ensures that our tests satisfy Definition 2.4, it does not ensure that our test rejects the null when it is false. To provide guarantees on the power of our test, we introduce sufficient conditions under which our tests for $\mathcal{H}(f)$ are tests of power one, i.e. rejects the null $\mathcal{H}(f)$ in finite time when $\tau$, the CMF/CATE of $P$, differs from $f$.

**Assumption 4.3** (Positive Covariance). Assume that there exists $\tau_\infty, v_\infty$ such that $\|\hat{\tau}_t - \tau_\infty\|_{L_2(P_X)} = o(1)$ and $\|\hat{v}_t - v_\infty\|_{L_2(P_X)} = o(1)$ almost surely, and $\exists c > 0$ such that

$$\mathbb{E}_{P_X}\left[\frac{(\tau_\infty(X) - f(X))(\tau(X) - f(X))}{v_\infty(X)}\right] \ge c. \quad (16)$$

Assumption 4.3 requires that sequence of sequential regressors $(\hat{\tau}_t, \hat{v}_t)_{t \in \mathbb{N}}$ has almost-sure $L_2(P_X)$ limits $\tau_\infty$ and $\hat{v}_t$, and the limiting weight function $(\tau_\infty(x) - f(x))/v_\infty(x)$ is positively correlated with the differences between the CMF/CATE $\tau$ and null $f$. Note that Assumption 4.3 requires $\tau(x) \ne f(x)$ in order to hold, implying that $P \notin \mathcal{H}(f)$, and does not require $g_t$ to converge in the setting of DGP2.

Leveraging Assumption 4.3, we formalize the conditions under which our test achieves power one in Theorem 4.4.

**Theorem 4.4** (Test of Power 1). *Let Assumptions 2.1- 2.3 and 4.1 hold. Then, for any distribution $P \notin \mathcal{H}(f)$ and regressors $(\hat{\tau}_t, \hat{v}_t)_{t \in \mathbb{N}}$ that satisfy Assumption 4.3,*

$$P\left(\exists t \in [t_0, \infty) : \xi_t(f, \rho, \alpha, t_0) = 1\right) = 1 \quad (17)$$

*for any fixed $\alpha \in (0, 1)$, $\rho > 0$, $l > 0$, $t_0 \in \mathbb{N}$, $f \in L_\infty(B)$, and sequence of weight magnitude bounds $(\epsilon_t)_{t \in \mathbb{N}}$.*

The results of Theorem 4.4 follow directly from Assumption 4.3. In particular, Assumption 4.3 ensures that the expected drifts of our process $\psi_t(f)$ are positive as $t \to \infty$, leading to almost-sure rejection and our test of power one result. Note that our tests *do not require* $\tau_\infty = \tau$ or $v_\infty$ to converge to the conditional variance of $\phi$ terms in order to achieve power one, ensuring rejection of the null under mild conditions.

Beyond rejection almost surely, our choice of weights for our test $\xi_t(f, \alpha, \rho, t_0)$ enables our testing procedure to obtain *optimal* sample complexities with respect to a Gaussian

location shift model. Below, we provide sufficient conditions under which our tests achieve this notion of optimality.

## 4.2. Achieving Optimal Sample Complexities

To achieve our desired sample complexities, we require our sequential regressors $\hat{\tau}_t$, $\hat{v}_t$, and $g_t$ converge almost surely in $L_2(P_X)$ to their corresponding values under $P$.

**Assumption 4.5** (Consistency). Assume that as $t \to \infty$, the (random) regression sequence processes $(\hat{\tau}_t, \hat{v}_t, g_t)_{t \in \mathbb{N}}$ satisfy the following almost surely: (i) $\|\hat{\tau}_t - \tau\|_{L_2(P_X)} = o(1)$, (ii) $\|g_t(\cdot, a) - \mu(\cdot, a)\|_{L_2(P_X)} = o(1)$ for $a \in \{0,1\}$ for DGP2, where $\mu(x,a)$ is as defined in Section 2, and (iii) $\|\hat{v}_t - v_*\|_{L_2(P_X)} = o(1)$, where $v_*(x) = \sigma^2(x)$ as in Equation (1) for DGP1, and $v_*(x) = \frac{\sigma^2(x,1)}{\pi(x,1)} + \frac{\sigma^2(x,0)}{\pi(x,0)}$ with $\sigma^2(x,a)$ and $\pi(x,a)$ defined in Equation (1) and Section 2 respectively for DGP2.

Assumption 4.5 requires our sequential regression functions $\hat{\tau}_t$, $g_t$, and $\hat{v}_t$ converge almost surely in $L_2(P_X)$ to their corresponding values under $P$. In particular, note that $v_*$ corresponds to the conditional variance $\mathbb{E}[(\phi_{t,*}^{CATE} - \tau(x))^2 | X = x]$ for DGP2, where $\phi_{t,*}$ is as defined in Equation (3) with $g_i = \mu$. To obtain expected sample complexity results, we rely on the following technical condition that

**Assumption 4.6** (Near Square Summability of Drifts). Let $\tau_\infty$ and $v_\infty$ be as defined in Assumption 4.3, and assume that there exists a function $g_\infty(\cdot, a)$ such that as $t \to \infty$, $\|g_t(\cdot, a) - g_\infty(\cdot, a)\|_{L_2(P_X)} = o(1)$ almost surely. Define the (random) drift as $d_t := \mathbb{E}_P[w_t(X, f)(\phi_t - f(X))]$, with $d_\infty$ being the corresponding limit drift with functions $\tau_\infty, v_\infty$ and $g_\infty$ in DGP2. Then, $\exists p > 1$, $\zeta \in (1/p, 1)$ such that the sum $\sum_{t=1}^\infty t^{p\zeta} \mathbb{E}_P[(d_t - d_\infty)^p]$ is finite.

Assumption 4.6 controls the deviations of drifts $d_t$ of our process $\psi_t(f)$ from its limit drift $d_\infty$ such that our almost sure limit and expected limit coincide. To interpret our condition, consider the case where $p = 2$: if $\zeta = 1/2$, then our condition is simply that expected squared deviations of drifts $d_t$ from $d_\infty$ are summable. Because we require $\zeta > 1/p$, this condition is a *slightly* stronger version of square summability for the deviations of the drift (Hall et al., 2014). We note that other works (Bibaut et al., 2024; Polyak & Juditsky, 1992) use similar conditions to establish limit results for expected convergence rates.

Using Assumptions 4.5 and 4.6, we establish asymptotic bounds (in the regime $\alpha \to 0$) on the sample complexity of our tests that match the optimal complexities for global null testing under a Gaussian shift model.

**Theorem 4.7** (Sample Complexities Bounds). *Let Assumptions 2.1- 2.3 and 4.1 hold. Let $t_0(\alpha)$ denote a sequence of burn-in times $\{t_0(\alpha)\}_{\alpha \in (0,1)}$ such that $t_0(\alpha) = o(\log(1/\alpha))$ and $\lim_{\alpha \to \infty} t_0(\alpha)$ diverges. De-*

*note $N_f(\alpha) = \inf\{t \geq t_0(\alpha) : \xi_t(\mathcal{H}, \alpha, \rho, t_0(\alpha)) = 1\}$ as the first time we reject the null $\mathcal{H}(f)$ for our $\alpha$-indexed sequence of tests $\xi_t(\mathcal{H}, \alpha, \rho, t_0(\alpha))$. Let $\Gamma(\tau, f)$ denote $\mathbb{E}_{P_X}\left[\frac{(\tau(X) - f(X))^2}{2\sigma^2(X)}\right]$ for DGP1, and*

$$\inf_{\mu_0 \in \mathcal{P}(f)} \mathbb{E}_{P_X}\left[\sum_{a \in \{0,1\}} \pi(X,a) \frac{(\mu(X,a) - \mu_0(X,a))^2}{2\sigma^2(X,a)}\right], \tag{18}$$

*in the setting of DGP2, with $\mathcal{P}(f)$ denoting the set of distributions $P$ with $\mu_0(x,a) = \mathbb{E}[Y | X = x, A = a]$ such that $\mu_0(x,1) - \mu_0(x,0) = f(x)$ $P_X$-almost surely.*

*For any distribution $P$ such that $\Gamma(\tau, f) > 0$ and sequential regressors $(\hat{\tau}_t, \hat{v}_t, g_t)_{t \in \mathbb{N}}$ such that Assumption 4.5 holds, for all fixed $\rho > 0$, $l > 0$, $\alpha \in (0,1)$, null function $f \in L_\infty(B)$, and sequences $(\epsilon_t)_{t \in \mathbb{N}}$,*

$$P\left(\limsup_{\alpha \to 0} \frac{N_f(\alpha)}{\log(1/\alpha)} \leq \Gamma^{-1}(\tau, f)\right) = 1 \tag{19}$$

*Furthermore, if Assumption 4.6 also holds, then we obtain*

$$\limsup_{\alpha \to 0} \frac{\mathbb{E}[N_f(\alpha)]}{\log(1/\alpha)} \leq \Gamma^{-1}(\tau, f), \tag{20}$$

*i.e. the expected sample complexity $\mathbb{E}[N_f]$ is upper bounded by the same constant as the almost sure limit as $\alpha \to 0$.*

Theorem 4.7 demonstrates that as error tolerance $\alpha \to 0$, the rejection time for $\mathcal{H}(f)$ is upper bounded almost surely and in expectation by the term $\Gamma^{-1}(\tau, f)$. In addition to our convergence assumptions, our result requires that $\alpha$-indexed tests $\xi_t(f, \rho, \alpha, t_0(\alpha))$ leverages burn-in times $t_0(\alpha) = o(\log(1/\alpha))$. The order of $t_0(\alpha)$ ensures that our burn-in times do not affect rejection time bounds, while allowing burn-in times $t_0(\alpha)$ to diverge to infinity.

*Remark* 4.8 (Optimality Bound for Gaussian Testing). The bound $\Gamma(\tau, f)$ corresponds to the *smallest* possible sample complexity for sequentially testing the null $\mathcal{H}(f)$, where outcomes $Y \sim N(\mu(x), \sigma^2(x)) | X = x$ and $Y \sim N(\mu(x,a), \sigma^2(x,a)) | X = x, A = a$ for DGP1 and DGP2 respectively with known conditional variance function $\sigma^2$ (Agrawal & Ramdas, 2025; Garivier & Kaufmann, 2016).[3] Under an asymptotic relaxation of error control and consistent nuisances, Theorem 4.7 demonstrates that our test recovers the optimal sample complexity for parametric Gaussian testing, even in highly nonparametric settings.

## 5. Experiments

To test our approach, we test the efficacy of our method on both synthetic and real world datasets. For all experiments, we set $\rho = 0.06$, $l = 0.01$, $\epsilon_t = t^{-0.24}/10$, and $\alpha = 0.1$,

---

[3]We provide a precise characterization in Appendix B.

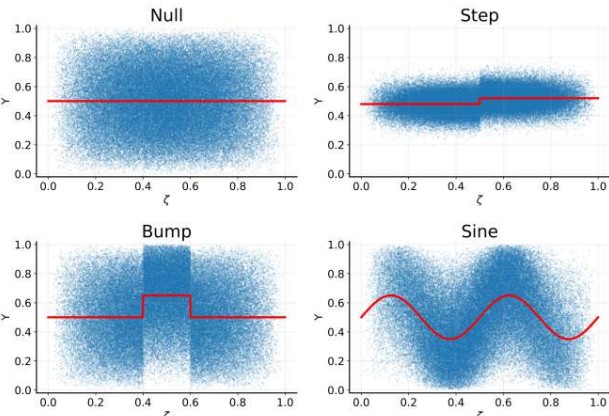

*Figure 2.* Scatterplot and conditional means (red lines) of $Y$ with respect to latent variable $\zeta$ for each of our synthetic examples.

where our choices satisfy the conditions of Theorem 4.2. We set the burn-in time $t_0$ as 250 and 2000 for our synthetic and real world datasets respectively.[4] For $\hat{\tau}_t, \hat{\sigma}_t^2, g_t$, we use an online variant of neural network (3 hidden layers) implemented in the `river` package in Python (Montiel et al., 2021). We provide additional details, including additional hyperparameter testing, in Appendix B.

## 5.1. Setup.

In our synthetic setups, we focus on testing the null $\mathcal{H}(f)$ in the setting of DGP1, where $f = 0.5$ is a constant function. For all synthetic examples, the context vector $X \in \mathbb{R}^{10}$ is generated with i.i.d. $X_i \sim \text{Unif}[0, 1]$. To generate outcomes, we sample $Y$ from a Beta distribution with conditional means $\tau(x) = f(\zeta)$, where $\zeta = \Phi\left(\frac{2}{\sqrt{10}} \sum_{i=1}^{10}(X_i - 0.5)\right)$, where $\Phi(\cdot)$ denotes the standard normal CDF. To assess error control and power against nonparametric alternatives, we test 4 synthetic conditional mean examples: (i) the null distribution, (ii) a step function, (iii) a bump function, and (iv) a sine curve. We visualize $Y$ with respect to the latent $\zeta$ in Figure 2.

For our real-world setup, we investigate the use of asymptotic AV inference for detecting CATE heterogeneity (e.g. $f(x) = 0$) with data from a randomized controlled trial (setting of DGP2). The French jobs dataset (Jobs) contains 33,797 observations of unemployed individuals participating in a large-scale randomized experiment comparing assistance programs (Behaghel et al., 2014). Our contexts $X \in \mathbb{R}^{41}$ contains both discrete and continuous demographic data for each individual, $Y$ corresponds to the (binary) outcome of reemployment in six months. The control group ($A = 0$) receives a privately-run counseling program, while the treatment group ($A = 1$) receives a publicly-run counseling program. To conduct multiple simulations, we

bootstrap 10,000 observations with population weights.

**Baseline Approaches.** As baselines for our approach, we include the asymptotic anytime valid $F$-test (Lindon et al., 2025) and Bonferroni-corrected asymptotic AV tests across conditions means under a discretized covariate space $\tilde{X}$ (Waudby-Smith et al., 2024; Chugg et al., 2025a) (Bin). Note that our latter approach mimics an asymptotic AV version of the max test (Arias-Castro et al., 2011). For our $F$-test baseline, we test the null that all coefficients (including intercept) are equal to zero for the synthetic setup. For the real-world setup, we test the null that the coefficients corresponding to the treatment indicator $A$ and all interaction terms $X \cdot A$ are equal to zero. For our synthetic setup, we bin based on the $L_2$ norm $\|X\|_2$ of context vectors and centered context vectors. For our real-world setup, we construct bins based on the age, education, and gender attributes of the context vector, constructing natural, interpretable bins for each conditional mean.

## 5.2. Discussion of Results.

To evaluate each method, we estimate the cumulative density function (CDF) for $N_f$, the first time in which we reject the null, up to time $T = 10,000$ (Figure 3). In the null plot, our CDFs demonstrate that all methods controls error uniformly at level $\alpha = 0.1$ throughout the horizon. Even with relatively small $t_0 = 250$, all methods provide AV error control, demonstrating asymptotic relaxations minimally affect error rates in practice.

In examples where $\tau \notin \mathcal{H}(f)$, our CDF plots characterize the power of our test. Among all approaches, only our method *consistently* achieves high rejection probability by $T = 10,000$. Among all tested methods, our approach achieves either the highest or second highest probability of rejecting the null across all examples uniformly over times $t \geq 3000$. The $F$-test baseline achieves high power when linear projections of $X$ capture variations in $\tau$ from $f$ (Step, Jobs), achieving rejection probabilities near one for sample sizes as small as $t = 1000$ in our Step example. However, its performance degrades rapidly when such projections fail to distinguish $\tau$ from $f$ (Bump, Sine), with rejection probabilities less than 0.2 by $T = 10,000$. By leveraging flexible regression estimates, our method avoids linear projection limitations and adapts to shape of $\tau$ across our examples.

Our binning baseline (e.g. Bonferroni correction across binned $X$) suffers from (i) fixed discretizations of the covariate space and (ii) conservative performance for large bin numbers due to multiple testing corrections. When the bins closely correspond to regions of the covariate space where either $\tau(x) > f(x)$ or $\tau(x) < f(x)$ uniformly over the bin (Bin(8) and Sine), our binning baseline outperforms all methods. However, when bins contain differing signals (e.g. $\tau(x) - f(x)$ differs within a bin), these tests can fail

---

[4]We discuss our choices for burn-in times $t_0$ in Appendix C.

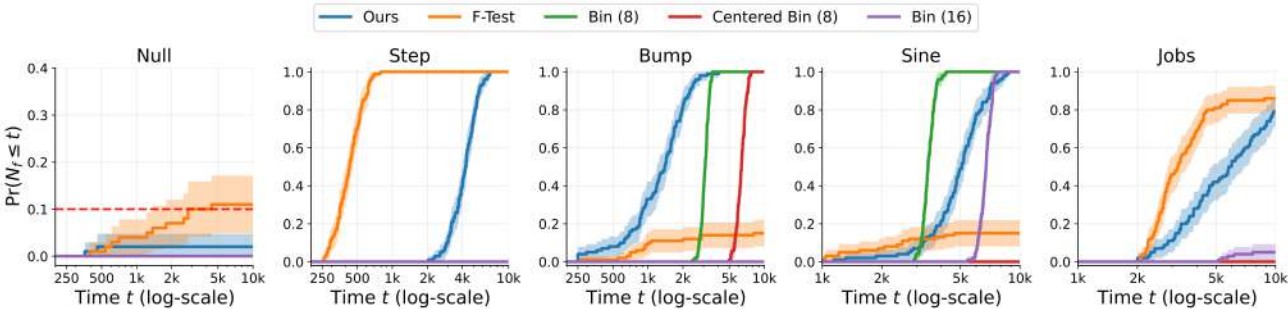

*Figure 3.* Cumulative density functions for the first time of rejection $N_f$. Shaded region denotes pointwise 95% confidence interval.

to reject the null completely (Step, Bin(8) in Jobs). While high fidelity bins provide multiple opportunities to capture regions of homogeneous $(\tau(x) - f(x))$ (Bin(16) in Jobs), they come at the cost of multiple testing corrections and smaller sample sizes per bin, reducing power. In contrast, our method avoids multiple testing corrections by tracking a single test statistic, and weights regions of the covariate space with adaptively estimated weights to maximize power.

## 6. Conclusions and Future Directions

This paper introduces GAAVI, a framework for asymptotic anytime-valid inference on global null hypotheses regarding conditional mean functions (CMF) and their contrasts (CATE). Our tests enable inference in fully nonparametric settings with general covariate spaces and allow for continuous monitoring beyond a sufficiently large burn-in time. Under simple boundedness conditions that do not require convergence of nuisance estimators, we establish asymptotic anytime-valid control of the type I error. We further provide mild regularity conditions under which our tests achieves power one and asymptotically optimal sample complexities relative to the parametric Gaussian shift problem.

The empirical studies support the theoretical results. Across a range of synthetic data-generating processes and a real-world randomized experiment, GAAVI maintains nominal error control under continuous monitoring while exhibiting competitive or superior power relative to existing asymptotic anytime-valid baselines. In particular, the results illustrate settings in which (i) methods based on linear projections or (ii) fixed discretizations of the covariate space incur substantial losses in power. In contrast, by permitting the use of flexible, ML-based regression estimators, our proposed testing procedure adapts to nonlinear or localized structure in the conditional mean function that may be poorly captured by linear projections or fixed discretizations.

Several directions for future work include relaxations of assumptions and extensions to additional methodological settings. We briefly outline some directions below.

**Relaxation of assumptions.** While the assumptions imposed in this work provide simple sufficient conditions for our theoretical guarantees, they may be relaxed. For example, the boundedness assumptions (Assumptions 2.2 and 4.1) could potentially be replaced by high-level moment conditions, as in Bibaut et al. (2024). In addition, the lower bound threshold $l$ imposed on the running variance estimates $\hat{v}_t$ may be allowed to decay toward zero, analogously to the lower bounds imposed on the weight magnitudes. We leave these technical extensions for future work.

**Time-varying distributions.** In many data collection settings, nonstationarity over time presents a significant practical challenge. Over the course of an experiment, the underlying data-generating distribution may evolve due to factors such as seasonality or response-adaptive behavior, violating our i.i.d. assumptions (Mineiro & Howard, 2023). In other cases, nonstationarity arises by design, as in experiments with time-varying or adaptive sampling schemes that depend on past observations (Bibaut & Kallus, 2024; Cho et al., 2025). We believe our proposed methodology can be extended to such settings, in a manner similar to Waudby-Smith et al. (2024), and leave a formal treatment of nonstationary regimes to future work.

**Composite testing procedures.** The proposed tests are designed for simple global null hypotheses. However, many hypotheses of practical interest, such as one-sided alternatives or comparisons across multiple treatments, correspond to composite nulls (Cho et al., 2024b; Cho & Kallus, 2025). While one could apply the proposed point-null tests for each hypothesis in the composite null, developing efficient sequential tests for composite null hypotheses on the CMF/CATE is an important promising direction for future research.

## Impact Statement

This paper presents work whose goal is to advance the field of Machine Learning. There are many potential societal consequences of our work, none which we feel must be specifically highlighted here.

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

# A. Appendix

Our appendix is organized as follows. First, in Appendix B, we provide additional remarks related to the main claim of the appendix, including (i) sufficient conditions for causal interpretation in DGP2, (ii) a precise definition for our notion of optimality in Theorem 4.7, and guidelines on our choice of tuning paramter $\rho$. In Appendix C, we provide additional details on our experiments and sensitivity analyses for our method regarding each hyperparameter. In Appendix D, we provide proofs for all theorems presented in the main body of our work.

# B. Additional Remarks

### B.1. Conditions for Causal Interpretation

To ensure DGP2 enables a causal interpretation (Imbens & Rubin, 2015), we provide a modified setup for DGP2 from the perspective of the Neyman-Rubin potential outcomes framework (Neyman, 1990; Rubin, 1974). We assume there exists an unobserved tuple $\tilde{O}_i = (X_i, Y_i(0), Y_i(1))$ at each time $i \in \mathbb{N}$, where $\tilde{O}_i$ is generated i.i.d. with respect to the joint distribution $\tilde{P}$. We refer to the tuple $(Y_i(0), Y_i(1))$ as *potential outcomes* for unit $i$ in the experiment. At each time $i \in \mathbb{N}$, the context $X_i \in \mathcal{X}$ is generated i.i.d. from the unknown marginal distribution $P_X$. After observing the context $X_i$, the experimenter assigns a binary treatment $A_i \in \{0, 1\}$, where $A_i$ is sampled according to a known policy $\pi(X_i, a) = P(A_i = a | X = X_i)$, corresponding to a randomized experiment. The outcome of interest $Y_i \in \mathbb{R}$ is then observed. Our observed tuple $O_i$ is then $(X_i, A_i, Y_i)$, corresponding to context/covariates $X_i$, treatment indicator $A_i$, and outcome $Y_i$ for unit $i$. On observed outcomes $Y_i$, we make the assumption commonly referred to as *consistency*.

**Assumption B.1** (Causal Consistency). The observed outcome $Y_i$ equals the potential outcome $Y_i(A_i)$.

Assumption B.1 ensures that the value of the treatment indicator $A_i$ does not refer to multiple versions of the treatment and avoids issues of interference/spillovers (e.g. the treatment of another unit $j \neq i$ affects the outcome $Y_i$ for unit $i$). We also make the following assumption, commonly referred to as *conditional exchangability*.

**Assumption B.2** (Conditional Exchangability). Treatment assignment is independent of potential outcomes given $X$, i.e. $(Y_i(0), Y_i(1)) \perp A_i | X_i$.

Assumption B.2 ensures that our experiment is properly randomized and rules out unmeasured confounding. In the setting where $X_i$ are pre-treatment outcomes (e.g. demographic data) and $\pi$ is chosen in advance by the experimenter, Assumption B.2 holds. Replacing our DGP2 setup with the setup presented above, the results of our work hold for the conditional average treatment effect (CATE) function defined as $\tau(x) = \mathbb{E}_P[Y(1) - Y(0)|X = x]$, rather than the CMF contrast definition $\tau(x) = \mathbb{E}[\mathbb{E}[Y|X = x, A = 1] - \mathbb{E}[Y|X = x, A = 0]|X = x]$ presented in Section 2.

### B.2. Optimality for Gaussian Shift Testing

In Theorem 4.7, we characterize the sample complexity of our approach as the error tolerance $\alpha$ shrinks to zero. In this section, we characterize the bounds $\Gamma(\tau, f)$ to provide a precise notion of optimality for our results. To do so, we first define two characteristics of a test: (i) anytime validity and (ii) power one.

**Definition B.3** (Anytime Valid Testing). Let $\mathcal{H}_0$ be a collection of distributions $P$, and let $\xi : H_\infty \times (0, 1) \to \{0, 1\}$ be a mapping from the observed data stream and nominal error level to a binary output. Then, the function $\xi : H_\infty \times (0, 1) \to \{0, 1\}$ is an $\alpha$-level anytime valid test for the null hypothesis $\mathcal{H}_0$ if $\forall P \in \mathcal{H}_0$, $P(\exists t \in \mathbb{N} : \xi(H_t, \alpha) = 1) \leq \alpha$.

Distinct from our definition of asymptotic anytime valid testing (Definition 2.4), an anytime valid test $\xi$ must satisfy time-uniform error control for all $t \in \mathbb{N}$, rather than all $t \geq t_0$. We note that this is a *stricter* requirement than that of Definition 2.4, and any test that satisfies Definition B.3 also satisfies Definition 2.4.

**Definition B.4** (Test of Power One). Let $\xi : H_\infty \times (0, 1) \to \{0, 1\}$ be an anytime valid test (Definition B.3) for the collection of distributions $\mathcal{H}_0$. Let $\mathcal{H}_1$ denote a set of distributions such as $\mathcal{H}_0 \cap \mathcal{H}_1 = \{\emptyset\}$, i.e. $\mathcal{H}_1$ is disjoint from the null set $\mathcal{H}_0$. Then, $\xi$ is a test of power one on the set of distribution $\mathcal{H}_1$ if for all $P \in \mathcal{H}_1$, $P(\exists t < \infty : \xi(H_t, \alpha) = 1) = 1$.

Definition B.4 states that the anytime valid test $\xi$ is a test of power one on distributions $\mathcal{H}_1$ if the test $\xi$ rejects the null $\mathcal{H}_0$ in finite time almost surely. To complete our setup, we now introduce the parametric setting for DGP1 and DGP2 that corresponds to our asymptotic upper bounds $\Gamma(\tau, f)$ in Definition 4.7.

**DGP1 (Parametric)** We assume $O_i = (X_i, Y_i)$ and $P = P_X \times P_{Y|X}$, where $X_i \in \mathcal{X}$ denotes the context (i.e. the conditioning variable) and $Y_i \in \mathbb{R}$ denotes the outcome of interest. We assume that outcomes $Y_i$ are generated according to a conditionally Gaussian model $Y_i \sim N(\tau(X_i), \sigma^2(X_i))$, where the variance function $\sigma^2 : \mathcal{X} \to \mathbb{R}_{++}$ is known in advance.

**DGP2 (Parametric)** We assume $O_i = (X_i, A_i, Y_i)$ and $P = P_X \times P_{A|X} \times P_{Y|A,X}$. For each observation, the context $X_i \in \mathcal{X}$ is generated i.i.d. from an unknown distribution $P_X$. After observing the context $X_i$, the experimenter assigns a binary treatment $A_i \in \{0, 1\}$, where $A_i$ is sampled according to a known policy $\pi(X_i, a) = P(A_i = a | X = X_i)$, corresponding to a randomized experiment. The outcome of interest $Y_i \in \mathbb{R}$ is then generated according to the conditional distribution $Y_i \sim N(\mu(X_i, A_i), \sigma^2(X_i, A_i))$, where the variance function $\sigma^2 : \mathcal{X} \times \{0, 1\} \to \mathbb{R}_{++}$ is known in advance.

In both cases, our DGPs are modified such that the outcome distributions of $Y$ are conditionally Gaussian, with known variances. The only unknown that is tested is the function $\tau$ (or equivalent functions that characterize $\tau$, such as $\mu(x, a)$ in the case of DGP2). Given the parametric setups above, we now establish that $\Gamma^{-1}(\tau, f)$ corresponds to the *optimal* sample complexity for the two parametric versions of DGP1 and DGP2 under our assumptions.

**Theorem B.5** (Asymptotic Lower Bounds for Parametric Model (Agrawal & Ramdas, 2025))**.** *Let $\mathcal{H}(f)$ denote the set of conditionally Gaussian distributions (as in DGP1 (Parametric) and DGP2 (Parametric)) with $\tau(x) = f(x)$, and let $\mathcal{H}(f')$ similarly denote the set of conditionally Gaussian distributions with CMF/CATE $\tau(x) = f'(x)$. Let $\Xi(\alpha) = \{\xi(\cdot, \alpha)\}$ denote the set of $\alpha$-level anytime valid tests (as in Definition B.3) with respect to the null $\mathcal{H}(f)$, and power one with respect to the alternative set $\mathcal{H}(f')$. Let $N_\xi(\alpha) = \inf\{t \in \mathbb{N} : \xi(H_t, \alpha) = 1\}$ denote the first time in which the test $\xi(H_t, \alpha)$ rejects the null $\mathcal{H}(f)$. Then, for any distribution $P \in \mathcal{H}(f')$, as $\alpha \to 0$, we obtain*

$$\lim_{\alpha \to 0} \inf_{\xi \in \Xi(\alpha)} \frac{\mathbb{E}_P\left[N_\xi(\alpha)\right]}{\log(1/\alpha)} \geq \Gamma^{-1}(f', f) \tag{21}$$

*where $\Gamma(\tau, f)$ is as defined in Theorem 4.7 for DGP1 and DGP2 respectively.*

Theorem B.5 states that for testing the parametric versions of DGP1 and DGP2 with known conditional variances, no anytime valid test with power one can have smaller expected sample complexity than $\Gamma^{-1}(\tau, f) \log(1/\alpha)$ as $\alpha \to 0$. Note that in Theorem 4.7, we establish that our test achieves this lower bound in our bounded, nonparametric setting under the assumption that nuisances converge (Assumption 4.5). Our results demonstrate that asymptotic relaxation of anytime validity (Definition 2.4) enable our tests in Section 3 to achieve the *optimal* asymptotic sampling complexity for standard anytime valid tests (Definition B.3) in the setting where (i) outcomes are assumed to be conditionally Gaussian, (ii) conditional variances are known, and (iii) the alternative hypothesis corresponds to the ground truth (i.e. the true DGP lies in $\mathcal{H}(f')$).

### B.3. Tuning of $\rho$ Parameter

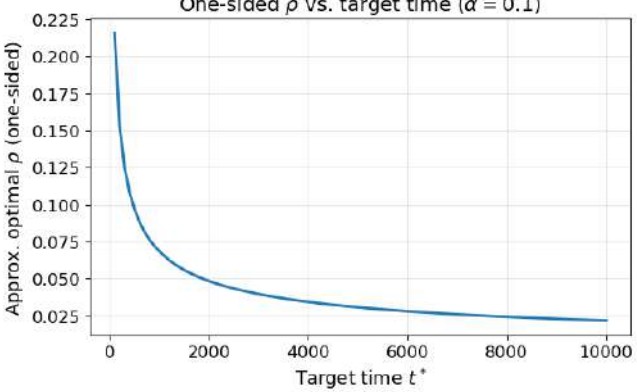

*Figure 4.* Plot of $\rho_{\text{1-sided}}$ for Differing values of $t^*$, $\alpha = 0.1$.

Following the choice of $\rho$ in Waudby-Smith et al. (2024), Cho & Kallus (2025), and (Howard et al., 2021), we provide the choice for $\rho$ that approximately yields the tightest bound at time $t^*$ in Equation (22):

$$\rho_{\text{1-sided}}(t^*, \alpha) = \sqrt{\frac{-2\log(2\alpha) + \log\big(-2\log(2\alpha) + 1\big)}{t^*}}. \tag{22}$$

To help visualize this function, we provide a plot for $\rho_{\text{1-sided}}(t^*, \alpha)$ in Figure 4. Larger values of $t^*$ result in smaller values of $\rho$, and as noted by Waudby-Smith et al. (2024), common ranges for $\rho$ are less than $0.1$. Our choice of $\rho = 0.06$ roughly corresponds to $t^* \approx 750$. We choose this value to ensure that the error budget $\alpha$ is spent during the horizon $T = 10,000$. To test the impact of $\rho$ on the results of our study, we refer readers to Figure 7, which tests various values of $\rho$ to assess its impact on rejection times.

### B.4. Choice of Burn-in Times

We choose differing burn-in times $t_0$ due to (i) ensuring valid error control in the real-world example and (ii) the differing number of covariates between the real-world and synthetic examples. As shown in our paper, asymptotic AV methods only approximately control error, and therefore larger burn-in times $t_0$ help ensure statistical validity of the analysis. We choose to set $t_0$ to be roughly 25 times $d$, the dimension of context set $\mathcal{X}$, in order to ensure that the linear regression routine in our $F$-test baseline (Lindon et al., 2025) stabilizes by the time in which we start monitoring. For our synthetic setup, where $d = 10$, we set $t_0 = 250$. In our real-world example, the linear regression model used in our $F$-test uses approximately 40 contextual variables and 40 interaction terms (covariate and treatment interaction), resulting in $d \approx 80$. As such, we set $t_0 = 2000$. We note that our choice of $t_0 = 25d$ is a simple heuristic used for our experiments; other works have suggested the rule of thumb for $t_0 = 10d$ in the setting of fixed-time inference for linear models (Harrell et al., 1996). For the effect of $t_0$ on the sample complexity of our method, we refer readers to Appendix C, where we conduct experiments with varying values of $t_0$ to assess its impact on (i) type I error and (ii) rejection times.

## C. Additional Experiment Results

We first provide relevant details that apply across all experiments. All experiments were run on a 14-inch, 2023 MacBook Pro with 16GB of RAM and an Apple M2 Pro chip. Our choice of sequential regressor for $\hat{\tau}_t$ and $\hat{v}_t$ is a neural network with 3 hidden layers, each with 64 dimensions. All hidden layer activation functions are the `ReLu` function, with the final layer using `Sigmoid` activation function. Note that by choosing the `Sigmoid` activation as our last layer, our sequential regressors $\hat{\tau}_t$ and $\hat{v}_t$ are bounded between zero and 1, satisfying Assumption 4.1. We use Adam($10^{-3}$) to optimize our regressor across all experiments, with all other parameters as the default parameters in the `river` package (Montiel et al., 2021). We set error tolerance $\alpha = 0.1$, predicted variance $\hat{v}_t$ lower bound $l = 0.01$, tightness parameter $\rho = 0.06$ (corresponding to $t^* \approx 750$). We set $t_0 = 250$ and $t_0 = 2000$ for synthetic and real-world experiments respectively.

In the remainder of this appendix, we provide (i) our synthetic data setup, (ii) our real-world data setup, and (iii) additional experiments that characterize the effect of hyperparameters on our method's performance.

### C.1. Synthetic Data Setup

To generate our four synthetic distributions (Null, Step, Bump, Sine), we set $X_i \in [0,1]^{10}$, where $X_{ij}$ denotes the $j$-th component of the vector $X_i$. For each $i \in \mathbb{N}$, $j \in [10]$, we generate i.i.d. $X_{ij} \sim \text{Unif}[0,1]$. For the conditional distribution $P(Y|X)$, we use a single-index latent variable $\zeta$ to construct Beta distributions that correspond to our desired conditional mean shape. Below, we specify our latent variable $\zeta$, and provide the conditional distributions for each synthetic setup.

**Latent Variable.** To construct our latent variable $\zeta$, we use the CDF of the standard normal distribution on a studentized version of the covariate/context $X_i$. Letting $\Phi(\cdot)$ denote the standard normal CDF, we define

$$\zeta_i = \Phi\left(\frac{2}{\sqrt{10}}\sum_{j=1}^{10}(X_{ij} - 0.5)\right) \tag{23}$$

as the latent corresponding to observation $i$. Note that $\zeta \in (0,1)$ due to being the output of the Gaussian CDF function.

**Conditional Distributions.** Our conditional mean functions take the form $m(\zeta)$, depending on only the latent variable $\zeta$.

$$m(\zeta) = 0.5 \tag{Null} \tag{24}$$
$$m(\zeta) = 0.5 + \delta \, \mathbf{1}[\zeta \geq 0.5] - \delta \, \mathbf{1}[\zeta < 0.5] \tag{Step} \tag{25}$$
$$m(\zeta) = 0.5 + \delta \, \mathbf{1}[0.4 \leq \zeta \leq 0.6] \tag{Bump} \tag{26}$$
$$m(\zeta) = 0.5 + \delta \, \sin(4\pi\zeta) \tag{Sine} \tag{27}$$

Using the conditional mean function $m(\zeta)$, we then generate outcomes $Y_i$ according to Beta distributions Beta$(c \, m(\zeta_i), c \, (1 - m(\zeta_i)))$, where $c$ denotes a scaling constant that does not affect the conditional mean. For our experiments, we set null with $c = 5$, bump with $\delta = 0.15$, $c = 5$, step with $\delta = 0.02$, $c = 50$, and sine with $\delta = 0.15$, $c = 5$. In Figure 5, we provide an enlarged version of Figure 2 in Section 5 to visualize our synthetic outcome distributions with respect to latent variable $\zeta$. Note that larger values of $c$ correspond to smaller conditional variances.

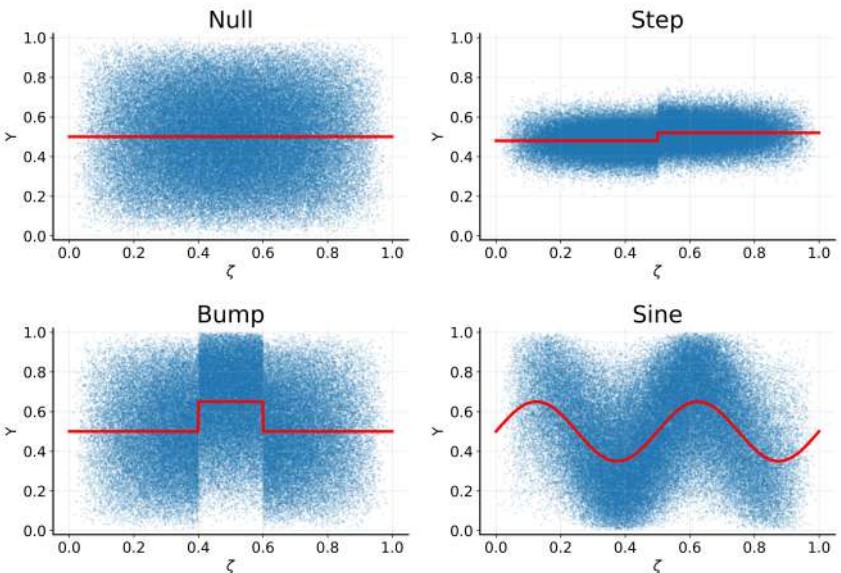

*Figure 5.* Enlarged version of Figure 2 in Section 5. Scatterplot and conditional means (red lines) of $Y$ with respect to latent variable $\zeta$ for each of our synthetic examples. Blue points denote 10,000 samples for each distribution.

### C.2. Real-World Setup

Using the data collected in Behaghel et al. (2014), we test our method on real-world data. In Behaghel et al. (2014), the authors analyze a large-scale randomized experiment comparing assistance programs offered to French unemployed individuals. In the original experiment, the authors compare three arms: (i) a "control" arm where individuals receive the standard public employment services, (ii) a "public" arm where individuals receive intensive counseling from a public agency, and (iii) a "private" arm where individuals receive intensive counseling from a private agency. The total experiment consists of 33,797 observations, with roughly 40 covariates/contexts per individual corresponding to demographic information.

Similar to the setup used in Kallus (2022); Cho et al. (2024c), we consider a hypothetical scenario where both private counseling and the control arm are designated as the control (e.g. $A = 0$) and the public counseling is designated as treatment (e.g. $A = 1$). This coincides with the hypothetical scenario where public employment services and private intensive counseling have both been available, and public intensive counseling corresponds to a new program that the French government considers introducing. As our outcome of interest $Y$, we use the binary outcome of reemployment in six months. Like Kallus (2022), we assume that the propensity scores are constant throughout the duration of the experiment and across contexts, where $\pi(x, 1) = 0.0923$, $\pi(x, 0) = 0.9077$ for all $x \in \mathcal{X}$.

**Setup for Dataset** To enable multiple simulated runs from the collected data, we use bootstrapped datasets with 10,000 observations, sampled with replacement according the population sample weights sw. For each observation $O_i = (X_i, A_i, Y_i)$, we set the context/covariate set $\mathcal{X} = \mathbb{R}^6 \times \{0, 1\}^{35}$, corresponding to 5 continuous variables and 35 binary indicators. Our

continous covariates/contexts are provided in line (28), and correspond to real-valued demographic data for each individual.

$$\texttt{age, Number\_of\_children, exper, salaire.num, mois\_saisie\_occ, ndem} \tag{28}$$

Our 35 binary covariates/contexts correspond to the pretreatment characteristics of each individual, including education, previous occupation, demographic data, region, unemployment reason, and statistical risk levels. We provide the names for each of these binary attributes in lines (29)-(39). For a full description of each covariate/context, we refer to Table 2 in Behaghel et al. (2014), which provides a full description of each variable in the study.

$$
\begin{array}{lr}
\texttt{College\_education, nivetude2, Vocational, High\_school\_dropout, Manager,} & (29) \\
\texttt{Technician, Skilled\_clerical\_worker, Unskilled\_clerical\_worker,} & (30) \\
\texttt{Skilled\_blue\_collar, Unskilled\_blue\_collar, Woman, Married,} & (31) \\
\texttt{French, African, Other\_Nationality, Paris\_region, North, Other\_regions,} & (32) \\
\texttt{Employment\_component\_level\_1, Employment\_component\_level\_2,} & (33) \\
\texttt{Employment\_component\_missing, Economic\_Layoff, Personnal\_Layoff,} & (34) \\
\texttt{End\_of\_Fixed\_Term\_Contract, End\_of\_Temporary\_Work,} & (35) \\
\texttt{Other\_reasons\_of\_unemployment, Statistical\_risk\_level\_2,} & (36) \\
\texttt{Statistical\_risk\_level\_3, Other\_Statistical\_risk,} & (37) \\
\texttt{Search\_for\_a\_full\_time\_position, Sensitive\_suburban\_area,} & (38) \\
\texttt{Insertion, Interim, Conseil} & (39)
\end{array}
$$

**Regression Functions for Pseudo-outcomes** Unlike our synthetic experiments, which were conducted under the setting of DGP1, our real-world example corresponds to the setting of DGP2, requiring predictable regressors $g_t(x, a)$ to construct terms $\phi_t$. For $g_t$, we train a sequential neural network model ($S$-learner style (Künzel et al., 2019)) using the `river` package (Montiel et al., 2021). For all experiments with real data, we use the 4 hidden dimensions of width 64. All hidden layer activations are the `ReLU` function, with the last layer as the `Sigmoid` function. We use the optimizer choice Adam($10^{-3}$), with all other parameters as the default setting.

**Details on Baselines** To accommodate the setting of DGP2, we modify our baselines to match the inference task $\mathcal{H}(0) : \tau(x) = 0$. For our $F$-test baseline (Lindon et al., 2025), we use the linear model

$$\mathbb{E}[Y|A, X] = c_0 + c_1 A + \beta_1 X + \beta_2 (A \cdot X), \tag{40}$$

where $A \cdot X$ denotes interaction terms, matching the setup of Section 3 in Lindon et al. (2025). For our $F$-test, we only include the coefficients $c_1 \in \mathbb{R}$, $\beta_2 \in \mathbb{R}^{|\mathcal{X}|}$ in order to test for treatment heterogeneity. Note that the coefficient $c_1$ denotes the average treatment effect, while $\beta_2$ corresponds to linear deviations from $c_1$ with respect to context set $\mathcal{X}$.

For our binning baseline, we use two-sided asymptotic AV confidence sequences from Waudby-Smith et al. (2024), with the hyperparameter $\rho = 0.06$ as our approach. As observations, we use the $\phi_t$ terms in DGP2 to form each confidence interval. To construct interpretable bins, we construct 8 bins using binary contexts `College_education`, `Woman`, and a binary discretizations of `age` using the median age of the dataset. For the 16 bin setup, we use the binary contexts `College_education`, `Woman`, and a discretizations of `age` by age quartiles of the dataset. For the variance used in the asymptotic AV confidence intervals, we use the sample variance of each bin's running mean of $\phi_t$, corresponding to the suggested approach in Waudby-Smith et al. (2024). We use $\alpha/b$ for each confidence sequence, where $b$ is the number of bins, and reject the null as soon as *any* of the confidence sequences do not include zero.

## C.3. Experiments under Alternative Setups

Our method involves multiple choices: (i) the tuning parameter $\rho$, (ii) the burn-in time $t_0$, (iii) the decay rate $\gamma$ that dictates the order of decay weight magnitude bounds $\epsilon_t \propto t^{-\gamma}$, (iv) the lower bound $l$ on conditional variance estimates $\hat{v}_t$, and (v) the choice of regression methods for predictable regressors $\hat{\tau}_t, \hat{v}_t$. To assess the impact of each parameter, we test various values of hyperparameters $t_0$, $\rho$, $\gamma$, and $l$ (Figure 7), as well as 4 different sequential regression methods for $\hat{\tau}_t$ (Figure 6). Before discussing our results, we provide the setup for our hyperparameter tuning experiments.

**Setup for Hyperparameter Experiments** For our hyperparameter $\rho$, $t_0$, $\gamma$, and $l$, we vary only the value of a single parameter for each row in Figure 7, with all other parameters fixed to the default values of $\rho = 0.06$, $t_0 = 250$, $\gamma = 0.24$,

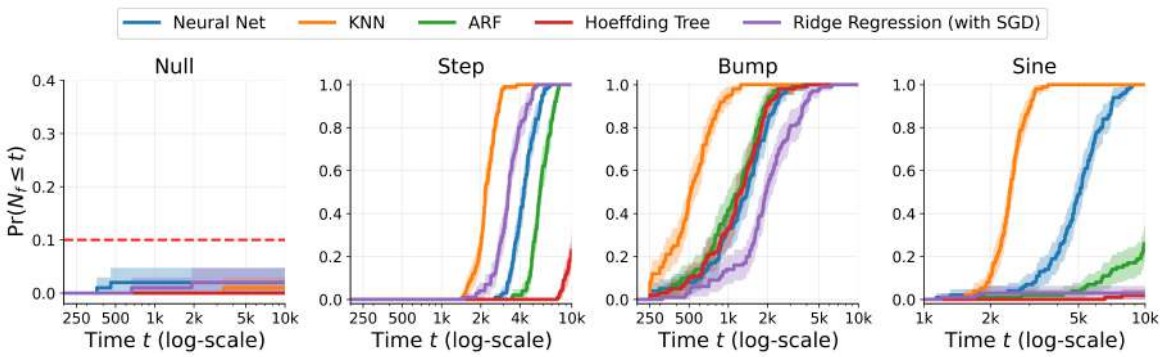

*Figure 6.* CDFs for $N_f$ under varying regression method choices for $\hat{\tau}_t$ with $t_0 = 250$, $l = 0.01$, $\rho = 0.06$, $\gamma = 0.24$ and $\hat{v}_t$ trained via neural network. Shaded region denotes pointwise 95% confidence interval.

and $l = 0.01$. For our regression testing experiments in Figure 6, we fix $(\rho, t_0, \gamma, l) = (0.06, 250, 0.24, 0.01)$, and vary the regression method for $\hat{\tau}$, with $\hat{v}_t$ using the neural network setup provided in the first paragraph of Section 5.

For our predictable regressors $\hat{\tau}_t$, we test four additional regression methods: (i) $k$-nearest neighbors, (ii) adaptive random forests (ARF) (Gomes et al., 2017), (iii) Hoeffding trees (Bifet et al., 2010), and (iv) ridge regression with stochastic gradient descent (SGD). For our $k$-NN baseline, we use $k = 50$ with the estimated prediction for $X_t$ weighted by the inverse $l_2$ distance between $X_t$ and $\{X_i\}_{i<t}$. For all other baselines, we use the `river` package, and initialize the regressors with

- ARF: `n_models = 30`, `seed = 0`.

- Hoeffding Tree: `grace_period = 250`, `leaf_prediction = "mean"`

- Ridge Regression w/ SGD: `optimizer = optim.SGD(0.01)`, `l_2 = 10^{-6}`.

and all other parameters as the default parameters for each method.

**Discussion of Regression Method.** Our power and sample complexity guarantees (Theorems 4.4, 4.7) rely on the behavior of our regression function $\hat{\tau}_t$ to achieve their respective results. Our experiments in Figure 6 reflect the impact of different regression methods on the distribution rejection times across our synthetic setups. For all methods, as guaranteed in Theorem 4.2, the type I error rate is strictly below the nominal error rate $\alpha = 0.1$, demonstrating our robust error guarantees. For the Step, Bump, Sine examples, we observe varying levels of performance. For flexible regression methods such as neural networks or $k$-NN, our tests reject the null with probability 1 by the end of the horizon $T = 10,000$; for poorly tuned regression methods (ARF, Hoeffding Tree) and restricted parametric models (Ridge Regression), our test demonstrates smaller probabilities for rejection. Like other methods that leverage predictable regressors (Waudby-Smith et al., 2024), we recommend ensembling prediction methods (Tsybakov, 2003; van der Laan et al., 2007) to achieve the best performance in practice. We leave the testing of ensembled predictors (e.g. $\hat{\tau}_t$ with a combination of all methods) for future work.

**Discussion of Hyperparameters.** Our additional experiments with hyperparameters $\rho, t_0, \gamma, l$ (shown in Figure 7) provide guidance on their selection. The parameters $t_0$ and $\gamma$ demonstrate that while theoretical results suggest large values of $t_0$ and $\gamma \in [0, 1/4)$, empirical performance (type I error, stopping probabilities) are roughly similar across a wide range of values. While our theory only applies for $t_0 \to \infty$, the results provided in the second row of Figure 7 demonstrate that type I error does not inflate even when $t_0 = 0$ over the horizon $T = 10,000$. Similarly, while our theory suggests $\gamma \in [0, 1/4)$ protects type I error, more aggressive rates of decay do not inflate type I error, demonstrating that $\gamma$ can be set larger than allowed when the weights $w_t(X_t, f)$ remain bounded away from zero as $t \to \infty$. Our results for the choice of $\rho$ and $l$ demonstrate that (i) $\rho$ should be set larger (corresponding to smaller desired $t^*$) to ensure that the error budget $\alpha$ is spent within the time horizon and (ii) $l$ should be set to a moderate value smaller than the magnitude of the estimated variance. Small values of $\rho$ (corresponding to a large value of $t^*$) result in smaller stopping probabilities for our synthetic examples. To ensure all testing budget $\alpha$ is spent before termination, we recommend setting $\rho$ (or $t*$) smaller than the expected time of rejection. For our lower bounds $l$, overly small values ($l = 0.005$) may degrade performance (Bump, Sine) due to the instability of $w_t(X_t, f)$, which uses the inverse variance estimate $\hat{v}_t$. However, overly large values of $l$ may prevent $\hat{v}_t$ from consistently estimating the desired conditional variance, degrading performance (Step with $l = 0.05, 0.1$). We leave adaptive tuning of the lower bound constant $l$ to future work.

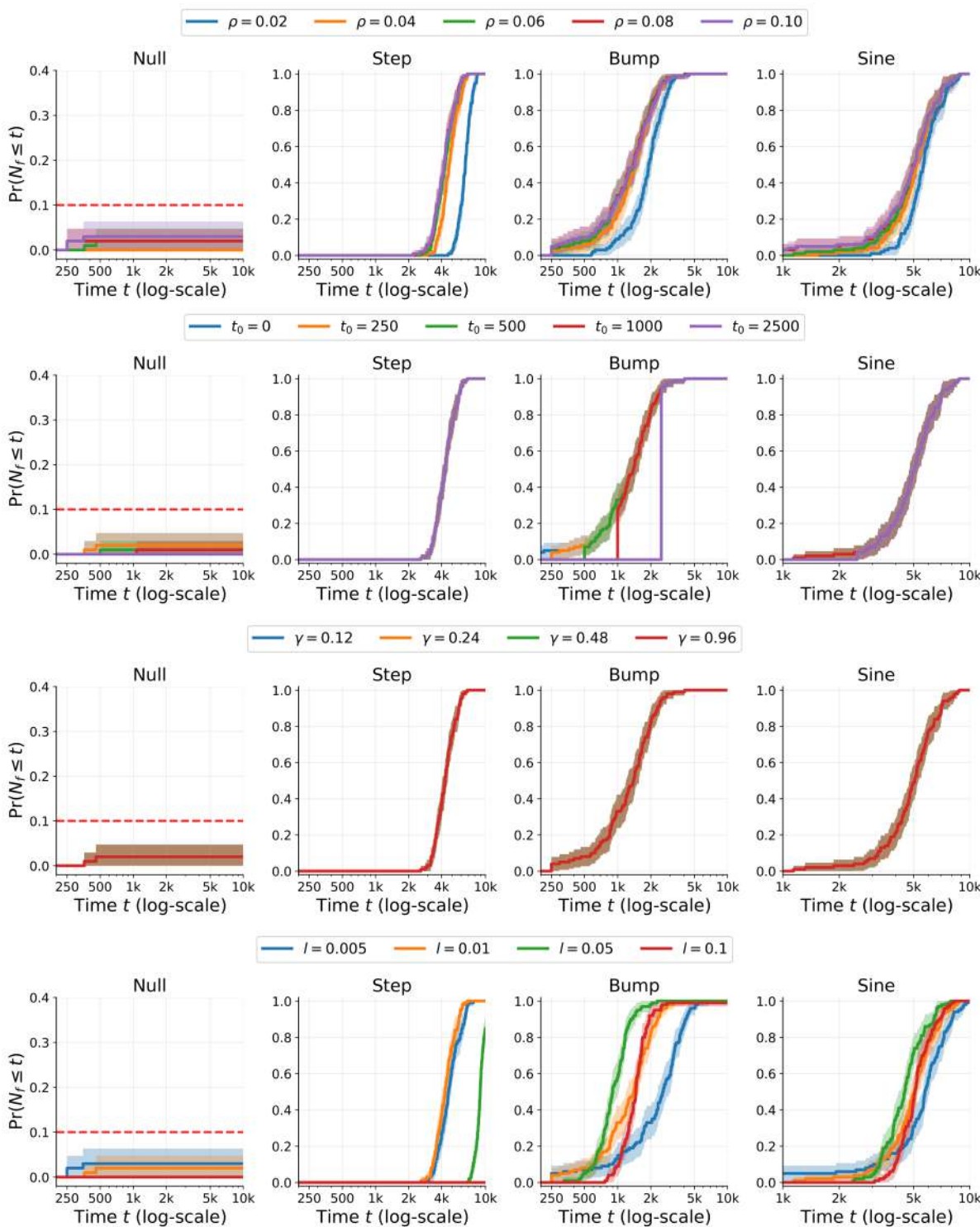

*Figure 7.* CDFs for $N_f$ under varying hyperparameter choices. Each row denotes changes to a single parameter, with all other parameters set as the default settings ($t_0 = 250$, $l = 0.01$, $\rho = 0.06$, $\gamma = 0.24$). Shaded region denotes pointwise 95% confidence interval.

# D. Proofs

Before starting, we provide preliminary lemmas from existing work used in our proofs. We prove our theorems in the order in which they appear, starting with our type I error guarantees in Theorem 4.2. When not stated explicitly, $\phi_t, \psi_t(f)$, and related terms refer to **both** DGP1 and DGP2, and either $\phi_t^{CMF}$ or $\phi_t^{CATE}$ satisfy the corresponding proof.

## D.1. Preliminary Lemmas for Proofs

To prove type I error guarantees, we rely heavily on an existing result obtained by (Waudby-Smith et al., 2024). Below, we provide a modified version of Theorem 2.8 in Waudby-Smith et al. (2024) that adapts their results to our setting.

**Lemma D.1** (Modified Theorem 2.8 of (Waudby-Smith et al., 2024)). *Let $(Z_t)_{t=1}^{\infty}$ be a sequence of random variables with conditional means $\mu_t := \mathbb{E}[Z_t|(Z_i)_{i=1}^{t-1}]$ and conditional variances $\sigma_t^2 := Var(Z_t|(Z_i)_{i=1}^{t-1})$. Let $(\mathcal{F}_t)_{t\in\mathbb{N}}$ denote the natural filtration, where $\mathcal{F}_t = \sigma((Z_i)_{i=1}^{t})$ and $\mathcal{F}_0$ is the trivial, empty sigma field. Let $\tilde{\sigma}_t^2$ be an estimator of cumulative variances $\frac{1}{t}\sum_{i=1}^{t} \sigma_t^2$. Assume that the following conditions (B1), (B2), and (B3) hold in an almost-sure sense:*

*(B1) Cumulative Variance Divergence: $\sum_{t=1}^{T} \sigma_t^2 \to \infty$,*

*(B2) Lyapnuov-Type Condition: $\exists \delta > 0$ such that $\sum_{t=1}^{\infty} \frac{\mathbb{E}\left[|Z_t - \mu_t|^{2+\delta}|(Z_i)_{i=1}^{t-1}\right]}{\left(\sum_{i=1}^{t}\sigma_i^2\right)^{1+\delta/2}} < \infty$,*

*(B3) Polynomial rate variance estimation: $\exists \eta \in (0,1)$ and predictable process $(\bar{v}_t)_{t\in\mathbb{N}}$ (i.e. $\bar{v}_t$ is $\mathcal{F}_{t-1}$-measurable) s.t. (i) $\bar{v}_t \geq \sigma_t^2$ for all $t \in \mathbb{N}$ and (ii) $\tilde{\sigma}_t^2 - \frac{1}{t}\sum_{i=1}^{t} \bar{v}_i = o\left(\frac{(\sum_{i=1}^{t} \bar{v}_i)^{\eta}}{t}\right)$.*

*Then, $\lim_{t_0 \to \infty} P\left(\exists t \geq t_0, \frac{1}{t}\sum_{i=1}^{t} \mu_i \leq \frac{1}{t}\sum_{i=1}^{t} Z_t - \sqrt{\frac{2(t\tilde{\sigma}_t^2 \rho_{t_0}^2 + 1)}{t^2 \rho_{t_0}^2} \log\left(\frac{\sqrt{t\tilde{\sigma}_t^2 \rho_{t_0}^2 + 1}}{2\alpha}\right)}\right) \leq \alpha.$*

This modified lemma differs from Theorem 2.8 of Waudby-Smith et al. (2024) only in the *variance estimation requirement*. In the original theorem, condition (B3) assumes that the variance estimator $\tilde{\sigma}_t^2$ tracks the *true* average cumulative conditional variance $\frac{1}{t}\sum_{i=1}^{t} \sigma_i^2$ at a polynomial rate, i.e. $\tilde{\sigma}_t^2 - \frac{1}{t}\sum_{i=1}^{t} \sigma_i^2 = o\left(\frac{\left(\sum_{i=1}^{t}\sigma_i^2\right)^{\eta}}{t}\right)$. In contrast, the modified statement introduces a predictable variance envelope $(\bar{v}_t)_{t\in\mathbb{N}}$ satisfying $\bar{v}_t \geq \sigma_t^2$ almost surely for all $t$, and replaces (B3) with a polynomial-rate approximation to the *upper-bound* average variance $\frac{1}{t}\sum_{i=1}^{t} \bar{v}_i$, i.e.$\tilde{\sigma}_t^2 - \frac{1}{t}\sum_{i=1}^{t} \bar{v}_i = o\left(\frac{\left(\sum_{i=1}^{t}\bar{v}_i\right)^{\eta}}{t}\right)$. Therefore, rather than requiring polynomial-rate estimation of the true vaverage cumulative conditional variance, the modified theorem only requires polynomial-rate tracking of a (possibly inflated) predictable upper bound. As a result, the resulting lower bound can be *nonsharp* (conservative) when $\sum_{i=1}^{t} \bar{v}_i$ is strictly larger than $\sum_{i=1}^{t} \sigma_i^2$.

We also leverage the martingale strong law of large numbers across our proofs to establish almost sure convergence.

**Lemma D.2** (Strong law for martingale differences (Theorem 2.18 of Hall et al. (2014))). *Let $(X_n, \mathcal{F}_n)_{n \geq 1}$ be a martingale difference sequence, i.e. $\mathbb{E}[X_n \mid \mathcal{F}_{n-1}] = 0$ a.s. for all $n$. Let $(U_n)_{n \geq 1}$ be a nondecreasing sequence of positive random variables such that $U_n$ is $\mathcal{F}_{n-1}$-measurable for each $n$. Fix $p \in (1, 2]$. Then, on the event $\left\{U_n \to \infty\right\} \cap \left\{\sum_{n=1}^{\infty} \frac{\mathbb{E}[|X_n|^p|\mathcal{F}_{n1}]}{U_n^p} < \infty\right\}, \frac{1}{U_n}\sum_{k=1}^{n} X_k \longrightarrow 0$ almost surely.*

To simplify convergence analysis of our running sums/averages, we use D.3.

**Lemma D.3** (Almost-sure convergence of Cesaro Means (Proposition 3 of Bibaut et al. (2020))). *If $t^{\beta} X_t \to 0$ almost surely, then for $\bar{X}_t := \frac{1}{t}\sum_{i=1}^{t} X_i$, $t^{\beta} \bar{X}_t \to 0$ almost surely.*

Lastly, to obtain the results of Theorem 4.7, we leverage Lemma D.4 in order to obtain the expected sample complexity.

**Lemma D.4** (Azuma's Inequality (Azuma, 1967)). *Let $(X_t)_{t\in\mathbb{N}}$ be a martingale with respect to filtration $(\mathcal{F}_t)_{t\in\mathbb{N}}$ and measure $P$. Assume that $X_t$ has bounded increments, i.e. $|X_t - X_{t-1}| \leq c$. Then, for all $n \in \mathbb{N}$, $\epsilon \in \mathbb{R}_+$,*

$$P(X_t - X_0 \leq -\epsilon) \leq \exp\left(\frac{-\epsilon^2}{2tc^2}\right). \tag{41}$$

## D.2. Proof of Theorem 4.2

To begin our proof, we first establish that our process $\psi_t(f)$ (as defined in Equation (4)) is a martingale for any distribution $P \in \mathcal{H}(f)$ (i.e. conditional mean zero). We then show that our martingale $\psi_t(f)$ satisfies the conditions of Lemma D.1. By applying Lemma D.1, we prove that our lower bounds $L_t(f, \rho, \alpha)$ cross above zero (the cumulative conditional mean) with probability at most $\alpha$ as $t_0 \to \infty$. By construction of our test $\xi_t$, our type I error guarantees follow immediately.

### D.2.1. CONDITIONAL MEAN ZERO

In the setting of DGP1, where $\phi_t = \phi_t^{CMF} = Y$, we obtain for every $P \in \mathcal{H}(f)$,

$$\mathbb{E}_P\left[\psi_t(f) - \psi_{t-1}(f)|\mathcal{F}_{t-1}\right] = \mathbb{E}_P\left[w_t(X_t, f)(\phi_t^{CMF} - f(X_t))|\mathcal{F}_{t-1}\right] \tag{42}$$

$$= \mathbb{E}_{P_X}\left[w_t(X_t, f)\left(\mathbb{E}_{P_{Y|X}}\left[\phi_t^{CMF}|X_t, \mathcal{F}_{t-1}\right] - f(X_t)\right)|\mathcal{F}_{t-1}\right] \tag{43}$$

$$= \mathbb{E}_{P_X}\left[w_t(X_t, f)\left(\mathbb{E}_{P_{Y|X}}\left[Y|X_t, \mathcal{F}_{t-1}\right] - f(X_t)\right)|\mathcal{F}_{t-1}\right] \tag{44}$$

$$= \mathbb{E}_{P_X}\left[w_t(X_t, f)\left(f(X_t) - f(X_t)\right)|\mathcal{F}_{t-1}\right] \tag{45}$$

$$= 0 \tag{46}$$

where line (43) follows from the fact that $w_t(X_t, f)$ is a $\mathcal{F}_{t-1}$-measurable function that only depends on $X_t$, line (45) follows from the fact that $P_{Y|X, \mathcal{F}_{t-1}} = P_Y$ by our assumptions on DGP1, and line (46) follows from the fact that $P \in \mathcal{H}(f)$, so $\mathbb{E}_{P_{Y|X}}[Y_t|X_t] = \tau(X_t) = f(X_t)$. Similarly, in the setting of DGP2, we also obtain that the martingale difference sequence corresponding to $\psi_t(f)$ has conditional means of zero for all $P \in \mathcal{H}(f)$. First, note that for all $P \in \mathcal{H}(f)$, for $\phi_{t,a}$ as defined in Section 3,

$$\mathbb{E}_P\left[\phi_{t,a}|X_t = x, \mathcal{F}_{t-1}\right] = \mathbb{E}_P\left[g_t(x, a) + \frac{\mathbf{1}[A_t = a](Y_t - g_t(x, a))}{\pi(x, a)}|X_t = x, \mathcal{F}_{t-1}\right] \tag{47}$$

$$= (1 - \pi_t(x, a))g_t(x, a) \tag{48}$$

$$+ \pi(x, a)\left(g_t(x, a) + \left(\mathbb{E}_P\left[\frac{Y_t - g_t(x, a)}{\pi(x, a)}\bigg|X_t = x, A_t = a, \mathcal{F}_{t-1}\right]\right)\right) \tag{49}$$

$$= g_t(x, a) + \pi(x, a)\left(\mathbb{E}_P\left[\frac{Y_t - g_t(x, a)}{\pi(x, a)}\bigg|X_t = x, A_t = a, \mathcal{F}_{t-1}\right]\right) \tag{50}$$

$$= g_t(x, a) + \mathbb{E}_P\left[Y|X = x, A = a\right] - g_t(x, a) \tag{51}$$

$$= \mathbb{E}_P\left[Y|X = x, A = a\right] \tag{52}$$

$$= \mu(x, a), \tag{53}$$

where $\mu(x, a)$ is as defined in the setup of DGP2 in Section 2, and $g_t$ can pulled out of conditional expectations $\mathbb{E}_P[\cdot|X_t = x, \mathcal{F}_{t-1}]$ due to being an $\mathcal{F}_{t-1}$-measurable function that depends only on $X_t$. By linearity of expectations, it follows that

$$\mathbb{E}_P\left[\phi_t^{CATE}|X_t = x, \mathcal{F}_{t-1}\right] = \mathbb{E}_P\left[\phi_{t,1}|X_t = x, \mathcal{F}_{t-1}\right] - \mathbb{E}_P\left[\phi_{t,0}|X_t = x, \mathcal{F}_{t-1}\right] = \mu(x, 1) - \mu(x, 0) = \tau(x). \tag{54}$$

By repeating the steps in lines (42)-(46) with $\phi_t^{CATE}$, we obtain that $\mathbb{E}_P\left[\psi_t(f) - \psi_{t-1}(f)|\mathcal{F}_{t-1}\right] = 0$ for all $P \in \mathcal{H}(f)$ in the setting of DGP2, showing that the difference sequence $(w_t(X_t, f)(\phi_t - f(X_t)))_{t \in \mathbb{N}}$ corresponding to $\psi_t(f)$ is indeed a martingale difference sequence with conditional means zero.

To proceed, we now show that the difference sequence corresponding to $\psi_t(f)$ satisfy Lemma D.1. To simplify notation, we set $Z_i := w_i(X_i, f)(\phi_i - f(X_i))$ as our martingale difference sequence terms, matching the notation of Lemma D.1.

### D.2.2. CUMULATIVE VARIANCE DIVERGENCE

Our cumulative variance divergence result is a direct consequence of (i) our boundedness assumptions (Assumptions 2.1, 2.2, 2.3, 4.1) and (ii) constraint on the weight magnitude bounds $(\epsilon_t)_{t \in \mathbb{N}}$ that state $\epsilon_t = \Omega(t^{-\gamma})$ for $\gamma \in [0, 1/4)$. Because $Z_t$ have conditional mean zero under $P \in \mathcal{H}(f)$, the conditional variance of $Z_t$, denoted as $\sigma_t^2$, is given by

$$\sigma_t^2 = \mathbb{E}_P\left[w_t^2(X_t, f)(\phi_t - f(X_t))^2|\mathcal{F}_{t-1}\right] = \mathbb{E}_P\left[w_t^2(X_t, f)\mathbb{E}_P\left[(\phi_t - f(X_t))^2|X_t, \mathcal{F}_{t-1}\right]|\mathcal{F}_{t-1}\right]. \tag{55}$$

In DGP1, plugging in $\phi_t^{CMF} = Y$, for some $c \in \mathbb{R}_{++}$ and $b$ as in Assumption 2.1, we obtain

$$\sigma_t^2 = \mathbb{E}_P\left[w_t^2(X_t, f)\mathbb{E}_P\left[(Y - f(X_t))^2|X_t, \mathcal{F}_{t-1}\right]|\mathcal{F}_{t-1}\right] = \mathbb{E}_{P_X}\left[w_t^2(X_t, f)\sigma^2(x)\right] \geq ct^{-2\gamma}b. \tag{56}$$

Likewise, in the setting of DGP2, plugging in $\phi_t^{CATE} = \phi_{t,1} - \phi_{t,0}$, we also obtain a similar result. The conditional variance of $(\phi_{t,1} - \phi_{t,0}) - f(x)$ for $P \in \mathcal{H}(f)$ is just the conditional variance of $(\phi_{t,1} - \phi_{t,0})$, which equals

$$\text{Var}_P\left((\phi_{t,1} - \phi_{t,0}) | X_t = x, \mathcal{F}_{t-1}\right) \tag{57}$$

$$= \text{Var}_P\left(g_t(x,1) + \frac{\mathbf{1}[A_t = 1](Y_t - g_t(x,1))}{\pi(x,1)} - \left(g_t(x,0) + \frac{\mathbf{1}[A_t = 0](Y_t - g_t(x,0))}{\pi(x,0)}\right) \bigg| X_t = x, \mathcal{F}_{t-1}\right) \tag{58}$$

$$= \text{Var}_P\left(\frac{\mathbf{1}[A_t = 1](Y_t - g_t(x,1))}{\pi(x,1)} - \left(\frac{\mathbf{1}[A_t = 0](Y_t - g_t(x,0))}{\pi(x,0)}\right) \bigg| X_t = x, \mathcal{F}_{t-1}\right) \tag{59}$$

Note that the mean of the term within the variance expression in line (59) is equal to

$$\mathbb{E}_P\left[\frac{\mathbf{1}[A_t = 1](Y_t - g_t(x,1))}{\pi(x,1)} - \left(\frac{\mathbf{1}[A_t = 0](Y_t - g_t(x,0))}{\pi(x,0)}\right) \bigg| X_t = x, \mathcal{F}_{t-1}\right] = f(x) - (g_t(x,1) - g_t(x,0)), \tag{60}$$

and thus the variance term in line (59) becomes

$$\text{Var}_P\left(\frac{\mathbf{1}[A_t = 1](Y_t - g_t(x,1))}{\pi(x,1)} - \left(\frac{\mathbf{1}[A_t = 0](Y_t - g_t(x,0))}{\pi(x,0)}\right) \bigg| X_t = x, \mathcal{F}_{t-1}\right) \tag{61}$$

$$= \mathbb{E}\left[\left(\frac{\mathbf{1}[A_t = 1](Y_t - g_t(x,1))}{\pi(x,1)} - \left(\frac{\mathbf{1}[A_t = 0](Y_t - g_t(x,0))}{\pi(x,0)}\right) - [f(x) - (g_t(x,1) - g_t(x,0))]\right)^2 | X_t = x, \mathcal{F}_{t-1}\right] \tag{62}$$

$$= \pi(x,1)\mathbb{E}\left[\left(\frac{Y_t - g_t(x,1)}{\pi(x,1)} - [f(x) - (g_t(x,1) - g_t(x,0))]\right)^2 | A_t = 1, X_t = x, \mathcal{F}_{t-1}\right] \tag{63}$$

$$+ \pi(x,0)\mathbb{E}\left[\left(\frac{Y_t - g_t(x,0)}{\pi(x,0)} + [f(x) - (g_t(x,1) - g_t(x,0))]\right)^2 | A_t = 1, X_t = x, \mathcal{F}_{t-1}\right] \tag{64}$$

We now reduce the terms on line (63). By adding and subtracting $\mu(x,1)$ to the numerator of $\frac{Y_t - g_t(x,1)}{\pi(x,1)}$, we obtain

$$\pi(x,1)\mathbb{E}\left[\left(\frac{Y_t - \mu(x,1)}{\pi(x,1)} + \frac{\mu(x,1) - g_t(x,1)}{\pi(x,1)} - [f(x) - (g_t(x,1) - g_t(x,0))]\right)^2 | A_t = 1, X_t = x, \mathcal{F}_{t-1}\right] \tag{65}$$

$$= \frac{\sigma^2(x,1)}{\pi(x,1)} + \pi(x,1)\left(\frac{\mu(x,1) - g_t(x,1)}{\pi(x,1)} - [f(x) - (g_t(x,1) - g_t(x,0))]\right)^2 \tag{66}$$

Repeating this step for the expression in line (64), we obtain a similar expression:

$$\pi(x,0)\mathbb{E}\left[\left(\frac{Y_t - \mu(x,0)}{\pi(x,0)} + \frac{\mu(x,0) - g_t(x,0)}{\pi(x,0)} + [f(x) - (g_t(x,1) - g_t(x,0))]\right)^2 | A_t = 1, X_t = x, \mathcal{F}_{t-1}\right] \tag{67}$$

$$= \frac{\sigma^2(x,0)}{\pi(x,0)} + \pi(x,0)\left(\frac{\mu(x,0) - g_t(x,0)}{\pi(x,0)} + [f(x) - (g_t(x,1) - g_t(x,0))]\right)^2 \tag{68}$$

Thus, our initial conditional variance expression is lower bounded by

$$\text{Var}_P\left((\phi_{t,1} - \phi_{t,0}) | X_t = x, \mathcal{F}_{t-1}\right) = \frac{\sigma^2(x,1)}{\pi(x,1)} + \pi(x,1)\left(\frac{\mu(x,1) - g_t(x,1)}{\pi(x,1)} - [f(x) - (g_t(x,1) - g_t(x,0))]\right)^2 \tag{69}$$

$$+ \frac{\sigma^2(x,1)}{\pi(x,1)} + \pi(x,1)\left(\frac{\mu(x,1) - g_t(x,1)}{\pi(x,1)} - [f(x) - (g_t(x,1) - g_t(x,0))]\right)^2 \tag{70}$$

$$\geq \frac{\sigma^2(x,1)}{\pi(x,1)} + \frac{\sigma^2(x,1)}{\pi(x,1)} \tag{71}$$

$$\geq 2b\kappa, \tag{72}$$

where the last line follows from Assumptions 2.3 and 2.1. Returning to our conditional variance term $\sigma_t^2$ for $Z_t$ in DGP2,

$$\sigma_t^2 = \mathbb{E}_P\left[w_t^2(X_t, f)\mathbb{E}\left[(\phi_{t,1} - \phi_{t,0} - f(X_t))^2 | X_t, \mathcal{F}_{t-1}\right] | \mathcal{F}_{t-1}\right] \geq ct^{-2\gamma}(2b\kappa) \tag{73}$$

for all $P \in \mathcal{H}(f)$ and some constant $c \in \mathbb{R}_{++}$. Thus, for DGP1 and DGP2, the conditional variance terms $\sigma_t^2$ lower bounded by terms of the order of $t^{-2\gamma}$, where $\gamma \in [0, 1/4)$. For any $\gamma \in [0, 1/4)$, note that $\lim_{t\to\infty} \sum_{i=1}^{t} \frac{1}{t^{2\gamma}}$ is infinite. Because our conditional variance terms are lower bounded on the order of $t^{-2\gamma}$, $\sum_{t=1}^{\infty} \sigma_t^2$ must diverge to infinity as well.

### D.2.3. LYAPUNOV-TYPE CONDITION

For our Lyapnov-Type condition, we require that $\exists \delta > 0$ such that $\sum_{t=1}^{\infty} \frac{\mathbb{E}\left[|Z_t|^{2+\delta}|\mathcal{F}_{t-1}\right]}{\left(\sum_{i=1}^{t}\sigma_i^2\right)^{1+\delta/2}} < \infty$. Before showing this result, we provide bounds on quantities of interest that allow for us to write a unified proof for DGP1 and DGP2:

- For both DGP1 and DGP2, $|\phi_t| \leq B^\phi := 2(B + 2B\kappa)$ for all $t \in \mathbb{N}$ under Assumptions 2.2, 2.3, and 4.1.

- For both DGP1, $\sigma_t^2 \geq b\mathbb{E}\left[w_t(X_t, f)^2\right]$ for all $t \in \mathbb{N}$, and for DGP2, $\sigma_t^2 \geq 2b\kappa\mathbb{E}\left[w_t(X_t, f)^2\right]$ for all $t \in \mathbb{N}$, as shown in our proof for cumulative variance divergence. To provide a unified treatment, we define $L_v := \min(b, 2b\kappa) = b > 0$.

- The weights $w_t(x, f)$ are bounded between $[ct^{-\gamma}, B/l]$ for all $t \in \mathbb{N}$, where $c \in \mathbb{R}_{++}$ is some fixed constant. The upper bound is a direct consequence of $l$, the conditional variance estimator lower bound, and Assumptions 4.1. As such, $\mathbb{E}\left[w_t^\delta(x, f)|\mathcal{F}_{t-1}\right] \in [c^\delta t^{-\delta\gamma}, B^\delta/l^\delta]$ for all $\delta \geq 0$.

We now provide a single proof for that both DGP1 and DGP2 satisfy the Lyapunov-type condition. We begin by lower bounding the cumulative variance term $\frac{1}{t}\sum_{i=1}^{t}\sigma_i^2$ for both DGP1 and DGP2:

$$\sum_{i=1}^{t}\sigma_i^2 \geq \sum_{i=1}^{t} L_v \mathbb{E}_{P_X}\left[w_i^2(X_t, f)|\mathcal{F}_{t-1}\right]. \tag{74}$$

Similarly, an upper bound for the numerator of the Lyapnuov condition for $P \in \mathcal{H}(f)$ is given by

$$\mathbb{E}\left[|w_i(X_i, f)|^{2+\delta}|\phi_i - f(X_i)|^{2+\delta}\Big|\mathcal{F}_{i-1}\right] \leq \left((B^\phi + B)B/l\right)^{2+\delta}, \tag{75}$$

which is a direct result from our bounds above and $f \in L_\infty(B)$. Putting this together, we obtain the upper bound

$$\sum_{t=1}^{\infty} \frac{\mathbb{E}\left[|Z_t|^{2+\delta}|\mathcal{F}_{t-1}\right]}{\left(\frac{1}{t}\sum_{i=1}^{t}\sigma_i^2\right)^{1+\delta/2}} \leq \sum_{t=1}^{\infty} \frac{\left((B^\phi + B)B/l\right)^{2+\delta}}{L_v^{1+\delta/2}\left(\sum_{i=1}^{t}\mathbb{E}_{P_X}\left[w_i^2(X_t, f)|\mathcal{F}_{t-1}\right]\right)^{1+\delta/2}} \tag{76}$$

$$= \frac{\left((B^\phi + B)B/l\right)^{2+\delta}}{L_v^{1+\delta/2}}\sum_{t=1}^{\infty} \frac{1}{\left(\sum_{i=1}^{t}\mathbb{E}_{P_X}\left[w_i^2(X_t, f)|\mathcal{F}_{t-1}\right]\right)^{1+\delta/2}} \tag{77}$$

$$\leq \frac{\left((B^\phi + B)B/l\right)^{2+\delta}}{L_v^{1+\delta/2}c^{2+\delta}}\sum_{t=1}^{\infty} \frac{1}{\left(t^{1-2\gamma}\right)^{1+\delta/2}} \tag{78}$$

where the inequality last line follows from our bounds on $w_t$. Ignoring the constant $\frac{\left((B^\phi + B)B/l\right)^{2+\delta}}{L_v^{1+\delta/2}c^{2+\delta}}$, our summation in line (78) is of the order $\sum_{t=1}^{\infty}\frac{1}{(t^{1-2\gamma})^{1+\delta/2}}$. In order for this sum to converge, we require $(1 - 2\gamma)(1 + \delta/2) > 1$. Rearranging our inequality, we obtain

$$(1 - 2\gamma)(1 + \delta/2) > 1 \iff \delta > 4\gamma/(1 - 2\gamma). \tag{79}$$

For any $\gamma \in [0, 1/4)$, there exists $\delta = 4\gamma/(1 - 2\gamma) + 1 > 0$ that satisfies this condition for both DGP1 and DGP2.

### D.2.4. Polynomial Variance Estimation

We now show that our running estimator $\hat{V}_t(f)$ satisfies condition $B3$. First, we define the predictable sequence $(\bar{v}_t)_{t\in\mathbb{N}}$ as

$$\bar{v}_t = \mathbb{E}_P\left[w_t^2(X_t, f)(\phi_t - \hat{\tau}_t(X_t))^2|\mathcal{F}_{t-1}\right] \tag{80}$$

$$= \mathbb{E}_P\left[w_t^2(X_t, f)\left(\phi_t - \tau(X_t) + \tau(X_t) - \hat{\tau}_t(X_t)\right)^2|\mathcal{F}_{t-1}\right] \tag{81}$$

$$= \mathbb{E}_P\left[w_t^2(X_t, f)\mathbb{E}_P[(\phi_t - \tau(x) + \tau(x) - \hat{\tau}_t(x))^2|X_t = x, \mathcal{F}_{t-1}]|\mathcal{F}_{t-1}\right] \tag{82}$$

$$= \underbrace{\mathbb{E}_P\left[w_t^2(X_t, f)\mathrm{Var}(\phi_t|X_t, \mathcal{F}_{t-1})\,\big|\,\mathcal{F}_{t-1}\right]}_{=\sigma_t^2} + \underbrace{\mathbb{E}_P\left[w_t^2(X_t, f)(\tau(X_t) - \hat{\tau}_t(X_t))^2|\mathcal{F}_{t-1}\right]}_{\geq 0} \tag{83}$$

For all distributions $P \in \mathcal{H}(f)$, note that the true conditional variance $\sigma_t^2$ of $Z_t$ is given by $\mathbb{E}_P\left[w_t^2(X_t, f)\mathrm{Var}(\phi_t|X_t, \mathcal{F}_{t-1})\,\big|\,\mathcal{F}_{t-1}\right]$, and therefore our terms $\bar{v}_t \geq \sigma_t^2$ almost surely for all $t \in \mathbb{N}$. Additionally, note that $\bar{v}_t$ is only a function of $\mathcal{F}_{t-1}$-measurable functions $w_t, \hat{\tau}_t$, and is therefore predictable.

To show that our variance converges at the desired polynomial rate, we first establish deterministic lower bounds on $\sum_{i=1}^t \bar{v}_i$. By our expression in line (83) and our bounds in Appendix D.2.3, note that

$$\sum_{i=1}^t \bar{v}_i \geq \sum_{i=1}^t i^{-2\gamma}L_v = \Theta(t^{1-2\gamma}). \tag{84}$$

Thus, our requirement is that there exists $\eta \in (0,1)$ such that $\hat{V}_t(f) - \frac{1}{t}\sum_{i=1}^t \bar{v}_i = o\left(\frac{t^{\eta(1-2\gamma)}}{t}\right) = o(t^{\eta-1-2\gamma\eta})$ almost surely. To prove this result, we leverage the strong law of large numbers for martingale difference sequences in Lemma D.2. First, we construct the martingale difference sequence $(\tilde{Z}_t)_{t\in\mathbb{N}}$, where

$$\tilde{Z}_t = w_t^2(X_t, f)(\phi_t - \hat{\tau}_t(X_t))^2 - \bar{v}_t. \tag{85}$$

Note that $\mathbb{E}_P[\tilde{Z}_t|\mathcal{F}_{t-1}] = 0$ by definition of $\bar{v}_t$. Additionally, note that by our bulleted bounds in Section D.2.3,

$$\left|w_t^2(X_t, f)(\phi_t - \hat{\tau}_t(X_t))^2\right| \leq \frac{4B^4}{l^2} \implies |\tilde{Z}_t| \leq C_{\tilde{Z}} := 8B^4/l^2 \tag{86}$$

i.e. the increments $\tilde{Z}_t$ are bounded. We now leverage Lemma D.2 to show that $\frac{1}{t^{0.5+\epsilon_1}}\sum_{i=1}^t \left(\tilde{Z}_t\right) \to 0$ almost surely, where $\epsilon_1 > 0$. Indexing with $t$, we set $U_t = t^{0.5+\epsilon_1}$ and $p = 2$. Note that

$$\sum_{t=1}^\infty \frac{\mathbb{E}_P\left[|\tilde{Z}_t|^2|\mathcal{F}_{t-1}\right]}{t^{1+2\epsilon_1}} \leq (8B^4/l^2)^2 \sum_{t=1}^\infty \frac{1}{t^{1+2\epsilon_1}} < \infty \tag{87}$$

for any $\epsilon_1 > 0$. Thus, by direct application of Lemma D.2, we obtain that

$$t^{0.5-\epsilon_1}\left(\frac{1}{t}\sum_{i=1}^t \tilde{Z}_t\right) = t^{0.5-\epsilon_1}\left(\hat{V}_t(f) - \frac{1}{t}\sum_{i=1}^t \bar{v}_i\right) \to 0 \tag{88}$$

almost surely, i.e. $\left(\hat{V}_t(f) - \frac{1}{t}\sum_{i=1}^t \bar{v}_i\right) = o(t^{-0.5+\epsilon_1})$ almost surely. We now show that for any $\gamma \in [0, 1/4)$, there exists choices of $\eta \in (0,1)$ and $\epsilon_1 > 0$ that satisfy $\eta - 1 - 2\gamma\eta = -0.5 + \epsilon_1$, satisfying our polynomial convergence rate. Defining $\epsilon_2 := 1/4 - \gamma$, note that $\epsilon_2 > 0$ by our bounds on $\gamma$. Fixing $\epsilon_2$, we obtain that our polynomial estimation rate requires

$$\eta(1 + 2\epsilon_2) - 1 = \epsilon_1 > 0. \tag{89}$$

For any $\epsilon_2 \in (0, 1/4]$ (equivalently, $\gamma \in [0, 1/4)$), note that we can pick $\eta = \frac{1+\epsilon_2}{1+2\epsilon_2} \in (0,1)$ such that $\epsilon_1 = \epsilon_2$. Thus, our variance estimate $\hat{V}_t(f)$ converges at desired polynomial rate to $\frac{1}{t}\sum_{i=1}^t \bar{v}_i$, ensuring our variance convergence condition.

### D.2.5. APPLYING LEMMA D.1

In our previous sections, we have shown that (i) $\psi_t(f)$ is a martingale under $\mathcal{H}(f)$, with mean-zero martingale difference sequences $Z_t = w_t(X_t, f)(\phi_t - f(X_t))$, and (ii) our process $\psi_t(f)$ satisfies the conditions of Lemma D.1. Then, by direct application of Lemma D.1, we obtain that our lower bounds $L_t(f, \alpha, \rho)$ in Equation (5) achieves

$$\lim_{t_0 \to \infty} P\left(\exists t \geq t_0 : L_t(f, \alpha, \rho) > 0\right) \leq \alpha. \tag{90}$$

for all distributions $P \in \mathcal{H}(f)$. Then, by definition, our testing procedure $\xi_t(f, \alpha, \rho, t_0)$ in Equation (7) achieves

$$\lim_{t_0 \to \infty} P\left(\exists t \geq t_0 : \xi_t(f, \alpha, \rho, t_0) = 1\right) = \lim_{t_0 \to \infty} P\left(\exists t \geq t_0 : L_t(f, \alpha, \rho) > 0\right) \leq \alpha, \tag{91}$$

recovering the guarantee of Equation (14) in Theorem 4.2. To obtain our confidence sequence guarantees, note that

$$P(\exists t \geq t_0 : \tau \notin C_t(\rho, \alpha, t_0)) = P(\exists t \geq t_0 : \xi_t(\tau, \alpha, \rho, t_0) = 1), \tag{92}$$

and therefore by Equation (91), we obtain the same guarantees, i.e. $\lim_{t_0 \to \infty} P\left(\exists t \geq t_0 : \tau \notin C_t(\rho, \alpha, t_0)\right) \leq \alpha$.

### D.3. Proof of Theorem 4.4

We now show that our test achieves power one for distributions $P \notin \mathcal{H}$ when Assumption 4.3 is satisfied. Our proof simply relies on the martingale strong law of large numbers presented in Lemma D.2 and the Cesaro convergence result in Lemma D.3. First, for $P \notin \mathcal{H}(f)$, we add and subtract the true CMF/CATE function $\tau$ to the terms in our process $\psi_t(f)$:

$$\psi_t(f) = \sum_{i=1}^{t} \left( \frac{(\hat{\tau}_i(X_i) - f(X_i))}{\hat{v}_i(X_i)} \right) (\phi_i - \tau(X_i) + (\tau(X_i) - f(X_i))) \tag{93}$$

$$= \underbrace{\sum_{i=1}^{t} \left( \frac{(\hat{\tau}_i(X_i) - f(X_i))}{\hat{v}_i(X_i)} \right) (\phi_i - \tau(X_i))}_{=\psi_t(\tau)} + \underbrace{\sum_{i=1}^{t} \left( \frac{(\hat{\tau}_i(X_i) - f(X_i))}{\hat{v}_i(X_i)} \right) (\tau(X_i) - f(X_i))}_{:=d_t} \tag{94}$$

$$= \psi_t(\tau) + d_t. \tag{95}$$

Under distribution $P$, which has CMF/CATE $\tau$, the process $\psi_t(\tau)$ is a martingale, with conditionally zero-mean martingale difference terms $Z_i = \left( \frac{(\hat{\tau}_i(X_i) - f(X_i))}{\hat{v}_i(X_i)} \right) (\phi_i - \tau(X_i))$. We now show that $t^{-1}\psi_t(\tau)$ converges almost surely to zero. Note that each term $Z_i$ is bounded as follows:

$$|Z_i| \leq \frac{2B}{l}(2B) = 4B^2/l, \tag{96}$$

where we assume that $|\tau(X_i)|$, the magnitude of the true CMF/CATE $\tau$, is bounded by $B$. Note that for the CATE example (DGP2), one can set $B$ to be twice the magnitude of the outcomes. It then follows that

$$\sum_{t=1}^{\infty} \frac{\mathbb{E}_P\left[|Z_t|^2 | \mathcal{F}_{t-1}\right]}{t^2} \leq (4B^2/l)^2 \sum_{t=1}^{\infty} \frac{1}{t^2} = (4B^2/l)^2 \frac{\pi^2}{6} < \infty. \tag{97}$$

By Lemma D.2, for $U_t = t$, $p = 2$, it follows that $\lim_{t \to \infty} t^{-1}\psi_t(\tau) = 0$ almost surely. We now show that the additional terms $d_t$ in line (95) converge to a constant $c > 0$ under Assumption 4.3 when normalized by $t^{-1}$. First, we center these additional terms to construct the martingale difference sequence

$$Z_i' = \left( \frac{(\hat{\tau}_i(X_i) - f(X_i))}{\hat{v}_i(X_i)} \right) (\tau(X_i) - f(X_i)) - \mathbb{E}_{P_X}\left[ \left( \frac{(\hat{\tau}_i(X) - f(X))}{\hat{v}_i(X)} \right) (\tau(X) - f(X)) \right] \tag{98}$$

Note that $\mathbb{E}[Z_i' | \mathcal{F}_{i-1}] = 0$ by construction. By the same argument as above with the martingale strong law of large numbers (e.g. finite summation with $U_t = t$, $p = 2$), we obtain that $\lim_{t \to \infty} t^{-1} \sum_{i=1}^{t} Z_i' = 0$ almost surely, i.e.

$$t^{-1}d_t - t^{-1}\sum_{i=1}^{t} \mathbb{E}_{P_X}\left[ \left( \frac{(\hat{\tau}_i(X) - f(X))}{\hat{v}_i(X)} \right) (\tau(X) - f(X)) \right] = o(1) \tag{99}$$

almost surely. We now show that averaged expectations $t^{-1} \sum_{i=1}^{t} \mathbb{E}_{P_X} \left[ \left( \frac{(\hat{\tau}_i(X) - f(X))}{\hat{v}_i(X)} \right) (\tau(X) - f(X)) \right]$ converge to our desired constant $c > 0$ using the Cesaro convergence result in Lemma D.3. With $\beta = 1$ in Lemma D.3, we only require that $\mathbb{E}_{P_X} \left[ \left( \frac{(\hat{\tau}_t(X) - f(X))}{\hat{v}_t(X)} \right) (\tau(X) - f(X)) \right]$ converges to $\mathbb{E}_{P_X} \left[ \left( \frac{(\tau_\infty(X) - f(X))}{v_\infty(X)} \right) (\tau(X) - f(X)) \right]$ almost surely, i.e.

$$ \mathbb{E}_{P_X} \left[ \left( \frac{(\hat{\tau}_t(X) - f(X))}{\hat{v}_t(X)} - \frac{(\tau_\infty(X) - f(X))}{v_\infty(X)} \right) (\tau(X) - f(X)) \right] = o(1) \tag{100} $$

almost surely under the assumption that $\|v_\infty - \hat{v}_t\|_{L_2(P_X)} = o(1)$ and $\|\tau_\infty - \hat{\tau}_t\|_{L_2(P_X)} = o(1)$. To show this, we split the expression within the expectation in line (100) into two distinct terms:

$$ \left( \frac{(\hat{\tau}_t(x) - f(x))}{\hat{v}_t(x)} - \frac{(\tau_\infty(x) - f(x))}{v_\infty(x)} \right) (\tau(x) - f(x)) = \underbrace{ \left( \frac{(\hat{\tau}_t(x) - f(x))}{\hat{v}_t(x)} - \frac{(\hat{\tau}_t(x) - f(x))}{v_\infty(x)} \right) (\tau(x) - f(x)) }_{=(a)} \tag{101} $$

$$ + \underbrace{ \left( \frac{(\hat{\tau}_t(x) - f(x))}{v_\infty(x)} - \frac{(\tau_\infty(x) - f(x))}{v_\infty(x)} \right) (\tau(x) - f(x)) }_{=(b)}. $$

$$ \tag{102} $$

We now show that terms $(a)$ and $(b)$ vanish via Cauchy Schwartz. Taking the expectation of term $(a)$, we obtain

$$ (a) = \mathbb{E}_{P_X} \left[ (v_\infty(X) - \hat{v}_t(X)) \frac{(\hat{\tau}_t(X) - f(X))(\tau(X) - f(X))}{\hat{v}_t(X) v_\infty(X)} \right] \le \frac{4B^2}{l^2} \|v_\infty - \hat{v}_t\|_{L_2(P_X)} = o(1) \tag{103} $$

almost surely by Assumption 4.3. Similarly, for term $(b)$, we obtain that the following holds almost surely:

$$ (b) = \mathbb{E}_{P_X} \left[ (\hat{\tau}_t(X) - \tau_\infty(X)) \frac{(\tau(X) - f(X))}{v_\infty(X)} \right] \le \frac{2B}{l} \|\hat{\tau}_t - \tau_\infty\|_{L_2(P_X)} = o(1). \tag{104} $$

Thus, by Lemma D.3, $t^{-1} \sum_{i=1}^{t} \mathbb{E}_{P_X} \left[ \left( \frac{(\hat{\tau}_i(X) - f(X))}{\hat{v}_i(X)} \right) (\tau(X) - f(X)) \right] - \mathbb{E}_{P_X} \left[ \left( \frac{(\tau_\infty(X) - f(X))}{v_\infty(X)} \right) (\tau(X) - f(X)) \right] = o(1)$ almost surely. Combining this result with the almost sure convergence in Equation (99), we obtain $t^{-1} d_t$ satisfies

$$ t^{-1} d_t - \mathbb{E}_{P_X} \left[ \left( \frac{(\tau_\infty(X) - f(X))}{v_\infty(X)} \right) (\tau(X) - f(X)) \right] = o(1). \tag{105} $$

Our power one results naturally follow from our almost sure convergence results. Our lower bound $L_t(f, \alpha, \rho)$ converges to

$$ L_t(f, \alpha, \rho) = t^{-1} \psi_t(f) - \ell_{t,\alpha,\rho}(\hat{V}_t(f)) \tag{106} $$

$$ = \underbrace{t^{-1} \psi_t(\tau)}_{=o(1)} + \underbrace{\mathbb{E}_{P_X} \left[ \left( \frac{(\tau_\infty(X) - f(X))}{v_\infty(X)} \right) (\tau(X) - f(X)) \right]}_{\ge c} + \underbrace{\ell_{t,\alpha,\rho}(\hat{V}_t(f))}_{=o(1)} \tag{107} $$

$$ \ge c \tag{108} $$

almost surely, where $\ell_{t,\alpha,\rho}(\hat{V}_t(f)) = \sqrt{ \frac{2(t\hat{V}_t(f)\rho^2 + 1)}{t^2 \rho^2} \log \left( 1 + \frac{\sqrt{t\hat{V}_t(f)\rho^2 + 1}}{2\alpha} \right) } = o(1)$ almost surely due to the boundedness of inputs $\hat{V}_t(f)$, $\rho$, and $\alpha$ and scaling on the order of $\Theta(\sqrt{\log t / t})$. Because the lower bound $L_t(f, \alpha, \rho) \ge c$ almost surely, it follows that for any fixed time $t_0$, $\lim_{t \to \infty} \xi_t(f, \alpha, \rho, t_0) = \lim_{t\infty} \mathbf{1}[\max_{t \in [t_0, t]} L_i(f, \alpha, \rho) > 0] = 1$ almost surely.

### D.4. Proof of Theorem 4.7

To prove Theorem 4.7, we first show that under Assumption 4.5, our normalized process $t^{-1} \psi_t(f) / \sqrt{V_t(f)}$ converges to $\sqrt{2\Gamma(\tau, f)}$ for both DGP1 and DGP2 almost surely. We begin with DGP1, and outline the key changes for DGP2. Note that for all parts of the proof, we assume $\Gamma(\tau, f)$ is strictly greater than zero; otherwise the bound is vacuous (i.e. infinite).

### D.4.1. CONVERGENCE OF NORMALIZED PROCESS FOR DGP1

Our results follow from the proofs of polynomial variance convergence in Appendix D.2.4 and Theorem 4.4 in Appendix D.3. As a reminder of the results, note that in Appendix D.2.4, we establish that $\hat{V}_t(f) - \frac{1}{t}\sum_{i=1}^{t}\bar{v}_i = o(1)$, where

$$\bar{v}_t = \mathbb{E}_P\left[w_t^2(X_t, f)\mathrm{Var}(\phi_t | X_t, \mathcal{F}_{t-1}) \,\middle|\, \mathcal{F}_{t-1}\right] + \mathbb{E}_P\left[w_t^2(X_t, f)(\tau(X_t) - \hat{\tau}_t(X_t))^2 | \mathcal{F}_{t-1}\right] \tag{109}$$

$$= \underbrace{\mathbb{E}_P\left[w_t^2(X_t, f)\sigma^2(X_t) \,\middle|\, \mathcal{F}_{t-1}\right]}_{=(a)} + \underbrace{\mathbb{E}_P\left[w_t^2(X_t, f)(\tau(X_t) - \hat{\tau}_t(X_t))^2 | \mathcal{F}_{t-1}\right]}_{(b)} \tag{110}$$

where the last line follows by $\phi_t = \phi_t^{CMF} = Y$ for the setting of DGP1. By deriving the almost sure limit of $\bar{v}_t$ under Assumption 4.5, we leverage the Cesaro convergence result in Lemma D.3 to find the limit of $\hat{V}_t(f)$. We begin with term $(b)$. By our bounds on $w_t$, note that $w_t^2(x, f) \leq (2B/l)^2$ for all $x \in \mathcal{X}$, $t \in \mathbb{N}$, $f \in L_\infty(B)$. Thus, term $(b)$ is bounded by

$$(b) \leq (2B/l)^2\|\hat{\tau}_t - \tau\|_{L_2(P_X)}^2, \tag{111}$$

where we drop conditioning on $\mathcal{F}_{t-1}$ due to the fact that $X_t \sim P_X$ i.i.d and $\hat{\tau}_t$ is predictable (i.e. we treat $w$ as a realization, and almost surely now applies to the random function $w$ instead of $\mathcal{F}_{t-1}$). Then, by the assumption that $\|\hat{\tau}_t - \tau\|_{L_2(P_X)} = o(1)$ almost surely, it follows that $\|\hat{\tau}_t - \tau\|_{L_2(P_X)}^2 = o(1)$ and term $(b) = o(1)$ almost surely. We now turn to term $(a)$. Defining the oracle weight $w_*(x, f) := \frac{\tau(x) - f(x)}{\sigma^2(x)}$,

$$\left|(a) - \mathbb{E}_{P_X}\left[w_*^2(X, f)\sigma^2(X)\right]\right| = \left|\mathbb{E}_{P_X}\left[\left(w_t^2(X, f) - w_*^2(X, f)\right)\sigma^2(X)\right]\right| \tag{112}$$

$$= \left|\mathbb{E}_{P_X}\left[\underbrace{\left(w_t(X, f) + w_*(X, f)\right)}_{\leq 4B/l}\left(w_t(X, f) - w_*(X, f)\right)\sigma^2(X)\right]\right| \tag{113}$$

$$\leq \frac{4B}{l}\mathbb{E}_{P_X}\left[|w_t(X, f) - w_*(X, f)|\,\sigma^2(X)\right] \tag{114}$$

$$\leq \frac{4B}{l}\|\sigma^2(\cdot)\|_{L_2(P_X)}\|w_t(\cdot, f) - w_*(\cdot, f)\|_{L_2(P_X)} \tag{115}$$

$$\leq \frac{4B^3}{l}\|w_t(\cdot, f) - w_*(\cdot, f)\|_{L_2(P_X)}, \tag{116}$$

where the first inequality follows from our bounds on $w_t$ and moving the abolsute value inside the expectation, the second inequality follows from Cauchy-Schwartz, and the last inequality follows from our outcome bounds (Assumption 2.2). We now show that $\|w_t(\cdot, f) - w_*(\cdot, f)\|_{L_2(P_X)}$ converges almost surely to zero under Assumption 4.5. Note that $w_t(x, f) - w_*(x, f)$ can be rewritten as

$$w_t(x, f) - w_*(x, f) = \frac{\hat{\tau}_t(x) - f(x)}{\hat{v}_t(x)} - \frac{\tau(x) - f(x)}{\sigma^2(x)} \tag{117}$$

$$= \frac{\hat{\tau}_t(x) - \tau(x)}{\hat{v}_t(x)} + \frac{\tau(x) - f(x)}{\hat{v}_t(x)\sigma^2(x)}(\sigma^2(x) - \hat{v}_t(x)), \tag{118}$$

and by subadditivity of the $L_2(P_X)$ norm, we obtain

$$\left|(a) - \mathbb{E}_{P_X}\left[w_*^2(X, f)\sigma^2(X)\right]\right| \leq \left\|\frac{\hat{\tau}_t - \tau}{\hat{v}_t}\right\|_{L_2(P_X)} + \left\|\frac{\tau - f}{\hat{v}_t\sigma^2}(\sigma^2 - \hat{v}_t)\right\|_{L_2(P_X)} \tag{119}$$

$$\leq l^{-1}\|\hat{\tau}_t - \tau\|_{L_2(P_X)} + \frac{2B}{lb}\|\sigma^2(x) - \hat{v}_t\|_{L_2(P_X)}, \tag{120}$$

where our second inequality is a consequence of Assumptions 4.3, 2.2, 4.1, and $f \in L_\infty(B)$. Because both norms are $o(1)$ almost surely under Assumption 4.5, it follows that $(a)$ converges to $\mathbb{E}_{P_X}[w_*^2(X, f)\sigma^2(X)]$ almost surely.

We now apply Lemma D.3 to establish the almost sure limit of $\hat{V}_t(f)$. By direct application of Lemma D.3, we obtain

$$\lim_{t \to \infty}\frac{1}{t}\sum_{i=1}^{t}\bar{v}_i = \lim_{t \to \infty}\bar{v}_t = \mathbb{E}_{P_X}\left[w_*^2(X, f)\sigma^2(X)\right] = \mathbb{E}_{P_X}\left[\left(\frac{\tau(X) - f(X)}{\sigma^2(X)}\right)^2\sigma^2(X)\right] = \mathbb{E}_{P_X}\left[\frac{(\tau(X) - f(X))^2}{\sigma^2(X)}\right],$$
$$\tag{121}$$

and by our previous results that establish $\hat{V}_t(f) - \frac{1}{t}\sum_{i=1}^t \bar{v}_i = o(1)$, $\hat{V}_t(f)$ converges to $\mathbb{E}_{P_X}\left[\frac{(\tau(X)-f(X))^2}{\sigma^2(X)}\right]$ almost surely.

We now derive the almost sure limit for $t^{-1}\psi_t(f)$. In the proof of Theorem 4.4, we establish that

$$\lim_{t\to\infty} t^{-1}\psi_t(f) = \underbrace{t^{-1}\psi_t(\tau)}_{=o(1)} + \mathbb{E}_{P_X}\left[\left(\frac{(\tau_\infty(X)-f(X))}{v_\infty(X)}\right)(\tau(X)-f(X))\right] \tag{122}$$

almost surely. Substituting $\tau = \tau_\infty$, $\sigma^2 = v_\infty$ (by Assumption 4.5), we obtain that $t^{-1}\psi_t(f)$ converges almost surely to $\mathbb{E}_{P_X}\left[\frac{(\tau(X)-f(X))^2}{\sigma^2(X)}\right]$, the same value as the limiting variance. By taking the ratio, we obtain that

$$\lim_{t\to\infty} \frac{t^{-1}\psi_t(f)}{\sqrt{\hat{V}_t(f)}} = \sqrt{\mathbb{E}_{P_X}\left[\frac{(\tau(X)-f(X))^2}{\sigma^2(X)}\right]} = \sqrt{2\Gamma(\tau,f)}, \tag{123}$$

almost surely, yielding the desired result.

### D.4.2. CONVERGENCE OF NORMALIZED PROCESS FOR DGP2

Leveraging the same steps as shown above, the assumption that $\|g_t(\cdot,a)\|_{L_2(P_X)} = o(1)$ almost surely for $a \in \{0,1\}$, and the variance decomposition provided in lines (69)-(70), we obtain the following bound for $t^{-1}\psi_t(f)/\sqrt{\hat{V}_t(f)}$:

$$\lim_{t\to\infty} \frac{t^{-1}\psi_t(f)}{\sqrt{\hat{V}_t(f)}} = \sqrt{\mathbb{E}_{P_X}\left[\frac{(\tau(X)-f(X))^2}{\frac{\sigma^2(X,1)}{\pi(X,1)} + \frac{\sigma^2(X,0)}{\pi(X,0)}}\right]}. \tag{124}$$

To convert this to our bound $\Gamma(\tau,f)$ for DGP2, we show matching upper and lower bounds for $\sqrt{\mathbb{E}_{P_X}\left[\frac{(\tau(X)-f(X))^2}{\frac{\sigma^2(X,1)}{\pi(X,1)} + \frac{\sigma^2(X,0)}{\pi(X,0)}}\right]}$

and $\sqrt{2\Gamma(\tau,f)}$, starting with $\sqrt{2\Gamma(\tau,f)} \geq \sqrt{\mathbb{E}_{P_X}\left[\frac{(\tau(X)-f(X))^2}{\frac{\sigma^2(X,1)}{\pi(X,1)} + \frac{\sigma^2(X,0)}{\pi(X,0)}}\right]}$. For any $\mu_0 \in \mathcal{P}(f)$, note that

$$\tau(x) - f(x) = (\mu(x,1) - \mu(x,0)) - (\mu_0(x,1) - \mu_0(x,0)) = (\mu(x,1) - \mu_0(x,1)) - (\mu(x,0) - \mu_0(x,0)). \tag{125}$$

For a fixed value of $x$, note that we can define $\tau(x) - f(x) = \boldsymbol{v}_1^\top \boldsymbol{v}_2$, where

$$\boldsymbol{v}_1 = \left(\frac{\sigma(x,1)}{\sqrt{\pi(x,1)}}, -\frac{\sigma(x,0)}{\sqrt{\pi(x,0)}}\right), \quad \boldsymbol{v}_2 = \left(\sqrt{\pi(x,1)}\frac{\mu(x,1)-\mu_0(x,1)}{\sigma(x,1)}, \sqrt{\pi(x,0)}\frac{\mu(x,0)-\mu_0(x,0)}{\sigma(x,0)}\right). \tag{126}$$

By applying Cauchy-Schwartz with respect to the $l_2$ norm for vectors $\boldsymbol{v}_1, \boldsymbol{v}_2$ for a fixed $x \in \mathcal{X}$, we obtain that

$$(\tau(x) - f(x))^2 \leq \left(\sum_{a\in\{0,1\}} \pi(x,a)\frac{(\mu(x,a)-\mu_0(x,a))^2}{\sigma^2(x,a)}\right)\left(\frac{\sigma^2(x,1)}{\pi(x,1)} + \frac{\sigma^2(x,0)}{\pi(x,0)}\right), \tag{127}$$

resulting in the inequality $\sum_{a\in\{0,1\}} \pi(x,a)\frac{(\mu(x,a)-\mu_0(x,a))^2}{\sigma^2(x,a)} \geq \frac{(\tau(x)-f(x))^2}{\frac{\sigma^2(x,1)}{\pi(x,1)} + \frac{\sigma^2(x,0)}{\pi(x,0)}}$. By taking the expectation of both sides, we

obtain for all $\mu_0 \in \mathcal{P}(f)$, $\mathbb{E}_{P_X}\left[\sum_{a\in\{0,1\}} \pi(X,a)\frac{(\mu(X,a)-\mu_0(X,a))^2}{\sigma^2(X,a)}\right] \geq \mathbb{E}_{P_X}\left[\frac{(\tau(X)-f(X))^2}{\frac{\sigma^2(X,1)}{\pi(X,1)} + \frac{\sigma^2(X,0)}{\pi(X,0)}}\right]$, showing that

$$\sqrt{2\Gamma(\tau,f)} = \sqrt{\inf_{\mu_0\in\mathcal{P}(f)} \mathbb{E}_{P_X}\left[\sum_{a\in\{0,1\}} \pi(X,a)\frac{(\mu(X,a)-\mu_0(X,a))^2}{\sigma^2(X,a)}\right]} \geq \sqrt{\mathbb{E}_{P_X}\left[\frac{(\tau(X)-f(X))^2}{\frac{\sigma^2(X,1)}{\pi(X,1)} + \frac{\sigma^2(X,0)}{\pi(X,0)}}\right]}. \tag{128}$$

To show equality, we provide the conditional mean function $\mu_0^* \in \mathcal{P}(f)$ that achieves the infimum in the line above. Denoting $\mu(x, a)$ as the true mean function for $P$ (as defined in Section 2), we define the function $\mu_0^*$ as follows:

$$\mu_0^\star(X, 1) = \mu(X, 1) - \left( \frac{\tau(X) - f(X)}{\frac{\sigma^2(X,1)}{\pi(X,1)} + \frac{\sigma^2(X,0)}{\pi(X,0)}} \right) \left( \frac{\sigma^2(X, 1)}{\pi(X, 1)} \right) \tag{129}$$

$$\mu_0^*(X, 0) = \mu(X, 0) + \left( \frac{\tau(X) - f(X)}{\frac{\sigma^2(X,1)}{\pi(X,1)} + \frac{\sigma^2(X,0)}{\pi(X,0)}} \right) \left( \frac{\sigma^2(X, 0)}{\pi(X, 0)} \right) \tag{130}$$

First, we show that $\mu_0^*$ is indeed in $\mathcal{P}(f)$. For every $x \in \mathcal{X}$, we obtain

$$\mu_0^\star(x, 1) - \mu_0^\star(x, 0) = \mu(x, 1) - \mu(x, 0) - \frac{\tau(x) - f(x)}{\frac{\sigma^2(x,1)}{\pi(x,1)} + \frac{\sigma^2(x,0)}{\pi(x,0)}} \left( \frac{\sigma^2(x, 1)}{\pi(x, 1)} + \frac{\sigma^2(x, 0)}{\pi(x, 0)} \right) \tag{131}$$

$$= \tau(x) - (\tau(x) - f(x)) = f(x). \tag{132}$$

Note that $\sum_{a \in \{0,1\}} \pi(X, a) \frac{(\mu(X,a) - \mu_0^*(X,a))^2}{\sigma^2(X,a)} = \frac{(\tau(X) - f(X))^2}{\frac{\sigma^2(X,1)}{\pi(X,1)} + \frac{\sigma^2(X,0)}{\pi(X,0)}}$. By taking expectations and square roots, we obtain

$$\sqrt{2\Gamma(\tau, f)} = \sqrt{\inf_{\mu_0 \in \mathcal{P}(f)} \mathbb{E}_{P_X} \left[ \sum_{a \in \{0,1\}} \pi(X, a) \frac{(\mu(X, a) - \mu_0(X, a))^2}{\sigma^2(X, a)} \right]} \tag{133}$$

$$\leq \sqrt{\mathbb{E}_{P_X} \left[ \sum_{a \in \{0,1\}} \pi(X, a) \frac{(\mu(X, a) - \mu_0^*(X, a))^2}{\sigma^2(X, a)} \right]} \tag{134}$$

$$= \sqrt{\mathbb{E}_{P_X} \left[ \frac{(\tau(X) - f(X))^2}{\frac{\sigma^2(X,1)}{\pi(X,1)} + \frac{\sigma^2(X,0)}{\pi(X,0)}} \right]}, \tag{135}$$

providing our desired result of equality.

### D.4.3. ALMOST SURE SAMPLE COMPLEXITY BOUNDS

We now leverage the almost sure limits of our normalized process $t^{-1}\psi_t(f)/\sqrt{V_t(f)}$ to provide almost sure bounds on the stopping time. We prove results for the setting where (i) $t_0(\alpha) \to \infty$ as $\alpha \to 0$ and (ii) $t_0(\alpha) = o(\log(1/\alpha))$ deterministically, matching the conditions of Theorem 4.7 in Section 4. We denote $N_f$ as the first time in which we reject the null for $t \in \mathbb{N}$, suppressing the dependence on $\alpha$ to simplify notation.

Note that at random time $N_f$, we must satisfy the following in order to stop for some $N_f > t_0(\alpha)$:

$$N_f \frac{\psi_{N_f}(f)}{N_f} - \sqrt{\hat{V}_{N_f}(f)} \sqrt{N_f \frac{2(\rho^2 + 1/(N_f \hat{V}_{N_f}(f)))}{\rho^2} \log \left( 1 + \frac{\sqrt{N_f \hat{V}_{N_f}(f)\rho^2 + 1}}{2\alpha} \right)} \in [0, c]. \tag{136}$$

The bound $c$ is a deterministic constant that upper bounds the overshoot (by boundedness of changes in $N_f L_t(f, \rho, \alpha)$) beyond zero and does not depend on $\alpha$. We rewrite this condition as the following for small enough $\alpha > 0$:

$$\left( N_f^{-1} \psi_{N_f}(f) / \sqrt{\hat{V}_{N_f}(f)} \right)^2 \frac{N_f}{\frac{2(\rho^2 + 1/(N_f \hat{V}_{N_f}(f)))}{\rho^2} \log \left( 1 + \frac{\sqrt{N_f \hat{V}_{N_f}(f)\rho^2 + 1}}{2\alpha} \right)} \tag{137}$$

$$\in \left[ 1, \left( 1 + \frac{c}{N_f \hat{V}_{N_f}(f) \frac{2(\rho^2 + 1/(N_f \hat{V}_{N_f}(f)))}{\rho^2} \log \left( 1 + \frac{\sqrt{N_f \hat{V}_{N_f}(f)\rho^2 + 1}}{2\alpha} \right)} \right)^2 \right]. \tag{138}$$

Because $t_0(\alpha) \to \infty$ as $\alpha \to 0$, $N_f \geq t_0(\alpha)$ deterministically, and $\hat{V}_{N_f}(f)$ converges to a positive constant larger than $b > 0$ (as shown in Appendices D.4.1, D.4.2, where $b$ is defined in Assumption 2.1), the rearrangement is valid (no dividing by 0) almost surely as $\alpha \to 0$.[5] Furthermore, note that as $\alpha \to 0$, by the convergence of $\psi_t(f)$ and $\hat{V}_t(f)$ to positive constants bounded away from zero almost surely (shown in Appendices D.4.1, D.4.2), it holds that there exists some constant $c \in \mathbb{R}_{++}$ such that $N_f \geq c \log(1/\alpha)$ as $\alpha \to 0$ almost surely for Equation (136) (and equivalently, Equation (137)) to hold.

Let $t(\alpha) = c \log(1/\alpha)$, and note that there exists an $\alpha' > 0$ such that $t_0(\alpha) < t(\alpha)$ for all $\alpha \leq \alpha'$. Therefore, the burn-in time $t_0(\alpha)$ does not interfere with $N_f$ for all $\alpha < \alpha'$ due to being deterministically smaller than the almost sure lower bound $t(\alpha)$ on rejection time $N_f$. Therefore, taking the limits on both sides with respect to $\alpha$, we obtain almost sure limits:

$$\lim_{\alpha \to 0} \left( N_f^{-1} \psi_{N_f}(f) / \sqrt{\hat{V}_{N_f}(f)} \right)^2 = 2\Gamma(\tau, f), \quad \lim_{\alpha \to 0} \frac{c}{N_f \hat{V}_{N_f}(f) \frac{2(\rho^2 + 1/(N_f \hat{V}_{N_f}(f)))}{\rho^2} \log \left( 1 + \frac{\sqrt{N_f \hat{V}_{N_f}(f) \rho^2 + 1}}{2\alpha} \right)} = 0,$$

(139)

where the first result follows from our previous derivations and the second result follows from the fact that $\hat{V}_{N_f}(f)$ converges to a constant greater than zero almost surely by Assumption 2.1. By the limits above, when $\Gamma(\tau, f) > 0$, we obtain

$$\lim_{\alpha \to 0} \frac{N_f}{\frac{2(\rho^2 + 1/(N_f \hat{V}_{N_f}(f)))}{\rho^2} \log \left( 1 + \frac{\sqrt{N_f \hat{V}_{N_f}(f) \rho^2 + 1}}{2\alpha} \right)} = \lim_{\alpha \to 0} \frac{N_f}{2 \log(1/\alpha)} \leq (2\Gamma(\tau, f))^{-1},$$

(140)

almost surely, and by multiplying both sides by 2, we obtain our desired result for the almost-sure upper bound.

### D.4.4. EXPECTED SAMPLE COMPLEXITY BOUNDS

We leverage our almost sure sample complexity results, along with Azuma's inequality (Lemma D.4), to establish that our expected asymptotic sample complexity is bounded by the same constant $\Gamma^{-1}(\tau, f)$. To begin, we first define

$$\beta_t(c, \alpha) := \sqrt{\frac{2(ct\rho^2 + 1)}{\rho^2} \log \left( 1 + \frac{\sqrt{tc\rho^2 + 1}}{2\alpha} \right)}$$

(141)

as the lower bound correction scaled by $t$ for $L_t(f, \rho, \alpha)$. Furthermore, let $t_\alpha(\epsilon) := (\Gamma^{-1}(\tau, f) + \epsilon) \log(1/\alpha)$, and let $d_t$ be the random drift as defined in Assumption 4.6. Then, by the tail sum identity for $\mathbb{E}_P[N_f]$, we bound the expectation as

$$\mathbb{E}_P[N_f] = \sum_{t=0}^{\infty} P(N_f > t)$$

(142)

$$= \sum_{t=t_0(\alpha)}^{t_\alpha(\epsilon)} \underbrace{P(N_f > t)}_{\leq 1} + \sum_{t=t_\alpha(\epsilon)}^{\infty} P(N_f > t)$$

(143)

$$\leq t_\alpha(\epsilon) + \sum_{t=t_\alpha(\epsilon)}^{\infty} P(N_f > t)$$

(144)

$$= (\Gamma^{-1}(\tau, f) + \epsilon) \log(1/\alpha) + \sum_{t=t_\alpha(\epsilon)}^{\infty} P(N_f > t),$$

(145)

where the last line follows from the definition of $t_\alpha(\epsilon)$. In the remainder of our proof, we provide bounds on the tail probability $\sum_{t=t_\alpha(\epsilon)}^{\infty} P(N_f > t)$, and send $\epsilon \to 0$ to show that our expected sample complexity bound holds.

---

[5]For a careful technical argument, one can construct a sample path argument and show that the set of sample paths where $\hat{V}_{N_f}$ remains close to zero has vanishing measure as $\alpha \to 0$. We omit this argument for the sake of brevity.

**Bounds on Cumulative Drift** We bound the tail probabilities $\sum_{t=t_\alpha(\epsilon)}^\infty P(N_f > t)$ by analyzing the sum $S_t :=$ $\sum_{i=1}^t d_i = t\Gamma^{-1}(\tau, f) + R_t$, where $R_t := \sum_{i=1}^t \left(d_i - \Gamma^{-1}(\tau, f)\right)$. By leveraging Assumption 4.6, there exists $p > 1$, $\zeta \in$ $(1/p, 1)$ such that $\sum_{t=1}^\infty t^{p\zeta} \mathbb{E}_P \left[(d_t - \Gamma^{-1}(\tau, f))^p\right] < \infty$. Let $Z := \left(\sum_{t=1}^\infty t^{p\zeta} \left(d_t - \Gamma^{-1}(\tau, f)\right)^p\right)^{1/p}$, with $\mathbb{E}[Z^p] < \infty$ by assumption. Furthermore, note that

$$t^\zeta \left|d_t - \Gamma^{-1}(\tau, f)\right| = \left(t^{p\zeta} \left|d_t - \Gamma^{-1}(\tau, f)\right|^p\right)^{1/p} \leq Z \implies \left|d_t - \Gamma^{-1}(\tau, f)\right| \leq Zt^{-\zeta}. \tag{146}$$

Then, our random variable $R_t$ is bounded as follows, where $c_1 \in \mathbb{R}_{++}$ is a fixed constant:

$$|R_t| \leq \sum_{i=1}^t \left|d_i - \Gamma^{-1}(\tau, f)\right| \leq Z \sum_{i=1}^t i^{-\zeta} \leq c_1 Z t^{1-\zeta}. \tag{147}$$

**Deterministic Margin Condition** In addition to bounds on $R_t$, note that $\Gamma^{-1}(\tau, f) > 0$ and $|\hat{V}_t(f)| \leq c_2$ for some constant $c_2 < \infty$ for all $t \in \mathbb{N}$, $f \in L_\infty(B)$ for both DGP1 and DGP2 by our boundedness assumptions. Then, note that there exists deterministic constant $d_\epsilon > 0$, $\alpha(\epsilon) \in (0, 1)$ such that for all $\alpha \leq \alpha(\epsilon)$, $t \geq t_\alpha(\epsilon)$,

$$t\Gamma^{-1}(\tau, f) - \beta_t(c_2, \alpha) \geq d_\epsilon t. \tag{148}$$

Note that $d_\epsilon$ and $\alpha(\epsilon)$ are completely deterministic (i.e. do not depend on sample paths). This follows immediately from the scaling of $\beta_t(c, \alpha)$ with respect to $t$ and $\alpha$, and the fact that $t \geq t_\alpha(\epsilon) = \Theta(\log(1/\alpha))$.

**Combining Bounds and Margins** To bound our tail probability terms $P(N_f > t)$ for $t \geq t_\alpha(\epsilon)$, note that

$$P(N_f > t) \leq P\left(\psi_t(f) \leq \beta_t(\hat{V}_t(f), \alpha)\right) \leq P\left(\psi_t(f) \leq \beta_t(c_2, \alpha)\right) \tag{149}$$

and for all $t \geq t_\alpha(\epsilon)$ and $\alpha \leq \alpha(\epsilon)$,

$$S_t - \beta_t(c_2, \alpha) = \sum_{i=1}^t d_i - \beta_t(c_2, \alpha) = \left(t\Gamma^{-1}(\tau, f) - \beta_t(c_2, \alpha)\right) + R_t \geq d_\epsilon t - |R_t|. \tag{150}$$

Now, define the sets $A_t := \left\{|R_t| \leq \frac{d_\epsilon t}{2}\right\}$ and $A_t^c = \Omega \setminus A_t$, where $P(\Omega) = 1$. Then, we bound $P(N_f > t)$ as

$$P(N_f > t) \leq P\left(\psi_t(f) \leq \beta_t(c_2, \alpha)\right) \tag{151}$$

$$= \underbrace{P\left(\psi_t(f) \leq \beta_t(c_2, \alpha),\ A_t\right)}_{:=T_1} + \underbrace{P\left(\psi_t(f) \leq \beta_t(c_2, \alpha),\ A_t^c\right)}_{:=T_2}. \tag{152}$$

For the term $T_1$, we use Azuma's inequality on the set of sample paths defined by $A_t$. To leverage Azuma's inequality, we analyze the event $\{\psi_t(f) \leq \beta_t(c_2, \alpha),\ A_t\}$. We rewrite $\psi_t(f) = S_t + M_t$, where $M_t = \sum_{i=1}^t \left(w_i(X_i, f)(\phi_i - f(X_i)) - d_i\right)$. Note that by definition, $\mathbb{E}\left[M_t - M_{t-1}|\mathcal{F}_{t-1}\right] = 0$, i.e. $M_t$ is a martingale. Furthermore, note that $|w_i(X_i, f)(\phi_i - X_i) - d_i| \leq c_3$ for all $i \in \mathbb{N}$, where $c_3 < \infty$ is a fixed constant that exists due our boundedness assumptions. Thus, the martingale $M_t$ has bounded increments. Then, on $A_t = \{|R_t| \leq d_\epsilon t/2\}$,

$$S_t - \beta_t(c_2, \alpha) \geq d_\epsilon t - |R_t| \geq d_\epsilon t - d_\epsilon t/2 = d_\epsilon t/2 \tag{153}$$

for $\alpha \leq \alpha(\epsilon)$, $t \geq t_\alpha(\epsilon)$ by Equations (148) and (150). Note that the event $\{\psi_t(f) \leq \beta_t(c_2, \alpha)\} = \{M_t \leq \beta_t(c_2, \alpha) - S_t\}$ by definition. When intersected with $A_t$, we obtain for all $\alpha \leq \alpha(\epsilon)$, $t \geq t_\alpha(\epsilon)$,

$$\{\psi_t(f) \leq \beta_t(c_2, \alpha),\ A_t\} \subseteq \left\{M_t \leq -\frac{d_\epsilon t}{2}\right\}. \tag{154}$$

Thus, we obtain $P\left(\psi_t(f) \leq \beta_t(c_2, \alpha),\ A_t\right)$ is upper bounded by

$$T_1 = P\left(\psi_t(f) \leq \beta_t(c_2, \alpha),\ A_t\right) \leq P\left(M_t \leq -\frac{d_\epsilon t}{2}\right) \leq \exp\left(\frac{-(d_\epsilon t/2)^2}{2tc_3^2}\right) = \exp\left(\frac{-td_\epsilon^2}{8c_3^2}\right) \tag{155}$$

where the second inequality above follows from Lemma D.4 (Azuma's inequality).

For term $T_2$, we bound the probability using our result from Equation (147). We can re-express the event $A_t^c$ as

$$A_t^c = \{|R_t| > d_\epsilon t/2\} \subseteq \{c_1 Z t^{1-\zeta} > d_\epsilon t/2\} = \left\{Z > \frac{d_\epsilon}{2c_1} t^\zeta\right\}. \tag{156}$$

By direct application of Markov's inequality, we obtain

$$T_2 \le P\left(A_t^c\right) \le P\left(Z > \frac{d_\epsilon}{2c_1} t^\zeta\right) = P\left(Z^p > \left(\frac{d_\epsilon}{2c_1}\right)^p t^{p\zeta}\right) \le \frac{\mathbb{E}\left[Z^p\right]}{\left(\frac{d_\epsilon}{2c_1}\right)^p t^{p\zeta}} \le c_4 t^{-p\zeta}, \tag{157}$$

where $c_4 < \infty$ is a fixed constant that exists due to $\mathbb{E}[Z^p] < \infty$ almost surely by Assumption 4.6.

**Summing over Probabilities**  Returning to Equation (145), we obtain the bound

$$\mathbb{E}_P\left[N_f\right] \le \left(\Gamma^{-1}(\tau, f) + \epsilon\right)\log(1/\alpha) + \sum_{t=t_\alpha(\epsilon)}^\infty P(N_f > t) \tag{158}$$

$$= \left(\Gamma^{-1}(\tau, f) + \epsilon\right)\log(1/\alpha) + \sum_{t=t_\alpha(\epsilon)}^\infty P(N_f > t, A_t) + \sum_{t=t_\alpha(\epsilon)}^\infty P(N_f > t, A_t^c) \tag{159}$$

$$= \left(\Gamma^{-1}(\tau, f) + \epsilon\right)\log(1/\alpha) + \sum_{t=t_\alpha(\epsilon)}^\infty \exp\left(\frac{-t d_\epsilon^2}{8 c_3^2}\right) + \sum_{t=t_\alpha(\epsilon)}^\infty c_4 t^{-p\zeta} \tag{160}$$

$$\le \left(\Gamma^{-1}(\tau, f) + \epsilon\right)\log(1/\alpha) + c_5 \exp\left(-t_\alpha(\epsilon)\right) + c_6 t_\alpha^{1-p\zeta}(\epsilon) \tag{161}$$

for some fixed constants $c_5, c_6 < \infty$. Plugging in $t_\alpha(\epsilon) = (\Gamma^{-1}(\tau, f) + \epsilon)\log(1/\alpha)$, and by the fact that $\zeta > 1/p$, it follows that $c_5 \exp\left(-t_\alpha(\epsilon)\right) + c_6 t_\alpha^{1-p\zeta}(\epsilon) = o(\log(1/\alpha))$. Taking the limit supremum of both sides with respect to $\alpha \to 0$, then sending $\epsilon \to 0$, we obtain our desired result in Theorem 4.7:

$$\limsup_{\alpha \to 0} \frac{\mathbb{E}_P[N_f]}{\log(1/\alpha)} \le \lim_{\epsilon \to 0}\left(\limsup_{\alpha \to 0} \frac{\left(\Gamma^{-1}(\tau, f) + \epsilon\right)\log(1/\alpha) + c_5 \exp\left(-t_\alpha(\epsilon)\right) + c_6 t_\alpha^{1-p\zeta}(\epsilon)}{\log(1/\alpha)}\right) = \Gamma^{-1}(\tau, f). \tag{162}$$

