# OpenReview forum: "GAAVI: Global Asymptotic Anytime Valid Inference for the Conditional Mean Function"
_ICML.cc/2026/Conference — ICML 2026 regular_

### Official Review · Reviewer_b2X7 · 2026-03-04

**Soundness:** 4
**Presentation:** 3
**Significance:** 3
**Originality:** 3
**Overall Recommendation:** 4
**Confidence:** 3

**Summary:**

This paper studies the sequential testing for conditional mean function in online data setting. The authors propose a nonparametric testing procedure using martingale based statistics to test the sequential data and provide any time valid asymptotic boundaries for sequential hypothesis testing. The author considered two different setting for the sequential data, generating CMF testing and CATE with binary treatments. The paper proves asymptotic anytime-valid type-I error control, power-one consistency, and optimal sample complexity relative to Gaussian shift models. Experiments on synthetic data and real-world training dataset demonstrate improved power compared to existing binning and F-test based baselines.

**Compliance With Llm Reviewing Policy:**

Affirmed.

**Key Questions For Authors:**

Q1. The proposed sequential statistic resembles a diffusion process under large-sample limits. Could the authors comment on how their method relates to or differs from diffusion/SDE-based analyses commonly used in classical sequential testing? A brief discussion positioning the method relative to this literature would help clarify the contribution.

Q2. It appears that Figure 3 corresponds to synthetic data experiments based on simulated DGPs. For the real-data analysis (Jobs dataset), the presentation is more limited and does not include comparable visualizations of sequential behavior. Providing similar plots for the real-data experiment would help readers better understand how the method behaves in practice.

**Limitations:**

Yes

**Strengths And Weaknesses:**

Strength:
1. This paper addresses an important problem in sequential testing for conditional mean functions, which is relevant for online experimentation, fairness auditing, and adaptive clinical trials.
2. This paper provides a coherent theory with asymptotic anytime-valid type-I error control, power-one consistency, and sample complexity optimality relative to Gaussian shift benchmarks.
3. The framework accommodates continuous/high-dimensional covariates and allows a wide range of nuisance estimators (e.g., k-NN, kernels, random forests, neural nets), making the approach broadly applicable beyond simple parametric regression.
4. Experiments on synthetic and real data suggest the approach can be more powerful than binning or linear-projection baselines, especially when the alternative is nonlinear and discretization loses signal.


Weakness:
1. The theoretical guarantees are asymptotic and requires certain amount of burn-in samples, what would the method perform when the sample size is limited.
2. The sequential conditional mean and variance are estimated on nuisance estimation. While the theory allows flexible estimators, the paper could better quantify sensitivity to nuisance error, especially under slow convergence or model misspecification.
3. Since the method is martingale-based, readers may naturally expect discussion of its relationship to modern e-value/e-process frameworks for anytime-valid testing, as well as classical diffusion/optimal stopping perspectives that characterize stopping times. A brief positioning section could improve clarity and context.

---

> ### Author Rebuttal · Authors · 2026-03-30
>
> We thank the reviewer for their detailed comments, suggestions, and questions. In particular, we appreciate that the reviewer finds our submission (i) addresses an important problem for real-world applications, (ii) theoretically coherent, and (iii) broadly application beyond parametric regression. Below, we hope to address the questions, comments, and clarifications raised by the reviewer below:
>
> **Limitations of Asymptotic Guarantees**: While our guarantees are indeed asymptotic, we note that (i) nonasymptotic guarantees for the same performance guarantees require significantly stronger modeling assumptions and (ii) error rates do not inflate empirically even for relatively small choices of $t_0$. Note that to obtain both exact error guarantees and the sample complexity results of Theorem 4.6, one needs to assume that data is generating according to an additive Gaussian noise model with known variances, a significantly stronger assumption than the setting of our work. For further details, we refer to rebuttal for Reviewer TGWx (Defining Optimal Sample Complexities). Our empirical results in Figure 6 (Appendix B, CDF of stopping time under different choices of $t_0$) show that type I error does not inflate even with very small burn-in times, suggesting that burn-in times can be relatively small in practice without inflating error rates. Thus, even in smaller sample regimes, our methods empirically tend to preserve type I error.
>
> **Nuisance Estimation Considerations**: Quantifying the effect of poor nuisance behavior is important, and we thank the reviewer for raising this point. Our theory supports slow rates of nuisance convergence for the guarantees of Theorem 4.4 and 4.6. In terms of model misspecification, the results of Theorem 4.6 generalize naturally. For example, in the setting of DGP1, our results generalize to an upper bound characterized by $\Gamma_\infty^{-1}(\tau, f)$, where $\Gamma_\infty(\tau, f)$ equals
>
> $$\Gamma_\infty(\tau, f) = \frac{\mathbb{E}\_{X}[w_\infty(X)(\tau_\infty(X) - f(X))]^2}{\mathbb{E}\_X[w_\infty^2(X)[\sigma^2(X) + (\tau(X) - \tau_\infty(X))^2]]}, $$
>
> with $w_\infty(x) = (\tau_\infty(x) - f(x))/v_\infty(x)$, and $\tau_\infty, v_\infty$ denoting the limiting nuisance functions. Note that $\Gamma_\infty$ is at most $\Gamma$ in Theorem 4.6, showing that misspecified nuisances lead to larger sample complexities (scales with $\Gamma^{-1}$). We have added the misspecified case for nuisances in our appendix, and a remark for the general case in Section 4.5.
>
> **In relation to e-processes**: While it is natural to compare our sequential testing procedure to modern e-value/process frameworks, asymptotic AV methods (including ours) are somewhat loosely connected: the process $\psi_t(f)$ is approximated via a Wiener process (random walk) beyond the burn-in time, and a mixture martingale (an e-process in the Gaussian setting based on integral of likelihood ratios) is leveraged to construct our lower bound test. Our test is not directly an e-process, and therefore power definitions such as GROW/e-power do not naturally apply. However, in the place of GROW/e-power, our work provides asymptotic sample complexity results, which are more standard for sequential testing (Garivier et al., 2016, Bibaut et al., 2024, Agrawal et al., 2025).
>
> To make this connection clear, we have added the statements above to Section 1.1 (related work).
>
> **Connection to Diffusions/Wiener Process**: As the reviewer notes, our test process does resemble a Wiener motion/diffusion process. Our error guarantees directly follow from approximating our test process $\psi_t(f)$ with a Wiener process, with $\hat{V}_t(f)$ serving as the intrinsic variance process. Our approach is common across asymptotic anytime valid testing literature (Waudby-Smith et al., 2024, Bibaut et al., 2024), who use the same invariance principles to ensure their error guarantees. While we state this connection in l. 275-280 (RHS), we have made this more explicit by adding a remark at the end of Theorem 4.2.
>
> **Real-World Experiment Results**: To clarify our experiment results in Figure 3, note that *all* plots shown correspond to CDF of the first rejection time under the same experiment settings (e.g. maximum sample size of 10,000 observations, same hyperparameters), with different underlying DGPs. The visualizations (e.g. cumulative probability of termination before $t$) across all synthetic and real-world dataset experiments are the same, with the same baseline methods plotted in each plot. While the "Jobs" plot appears to have less results (e.g. less baselines plotted) than our synthetic results, this is not the case - baselines other than the F-test and Bin (16) lack detection power on this dataset. Similar phenomena is shown in the "Step" example, where only the F-test baseline is well-powered. If the reviewer is referring to another aspect of Jobs in Figure 3, please let us know, and we would be happy to answer additional questions!

---

> > ### Author Rebuttal · Reviewer_b2X7 · 2026-04-04
> >
> > Thank your for the detailed rebuttal! I hope you can make the final version better using these justifications. I will not raise my score at this time and I wish you good luck!

---

### Official Review · Reviewer_TGWx · 2026-03-11

**Soundness:** 3
**Presentation:** 3
**Significance:** 3
**Originality:** 3
**Overall Recommendation:** 4
**Confidence:** 3

**Summary:**

This paper considers the setting of developing anytime valid asymptotically valid sequential tests for the conditional mean function (CMF) and conditional average treatment effect (CATE) in a unified framework. CMF and CATE are highly relevant objects of interest. The key contribution is to develop first-of-a-kind results for anytime valid asymptotically valid sequential tests for testing the CMF during data collection when covariates/contexts are continuous or high-dimensional. These tests are designed to test the Global Null. A mixture martingale method is developed that leverages flexible regression methods to maximize power against the specified global null.

**Compliance With Llm Reviewing Policy:**

Affirmed.

**Final Justification:**

My questions have been adequately addressed and I am maintaining my score.

**Key Questions For Authors:**

Apart from the questions raised in Strength and Weakness section, I found Remark 4.7 somewhat difficult to follow, and would greatly appreciate it if the authors could provide a more detailed explanation of its purpose and significance.

**Limitations:**

Yes

**Strengths And Weaknesses:**

### Presentation and Soundness
The paper is well-written and the results are backed by theoretical support. However, there is one concern regarding the use of the optimal sample complexities (see lines 61--62, left column, and used at other places as well). I am not sure what the authors mean by optimal sample complexities without describing any lower bound analysis on the sample complexity. I would like the authors to clarify this.

### Significance
I understand that the motivation for studying the general CMF inference task is very relevant in practice.

### Originality
I found the problem to be original. I would like to understand the technical contribution of the paper from the authors and how much of it uses (Waudby-Smith et al., 2024).

---

> ### Author Rebuttal · Authors · 2026-03-30
>
> We thank the reviewer for their detailed comments and questions. In particular, we appreciate that the reviewer finds our submission (i) well-written, (ii) theoretically well-supported, and (iii) first-of-a-kind for asymptotic anytime valid sequential testing of the CMF. Below, we hope to address the questions, comments, and clarifications raised by the reviewer below:
>
> **Defining Optimal Sample Complexities**: We agree with the reviewer that our definition of optimal sample complexities may be unclear as written. In this work, we mean optimal sample complexities as achieving the lower bound for asymptotic sample complexities under a Gaussian model under exact anytime valid guarantees. To be specific, for DGP1, consider the DGP $Y_i \sim \tau(X_i) + \epsilon_i$, where $\epsilon_i \sim N(0, \sigma^2(X_i))$ and the function $\sigma^2$ is known. Let $\Xi$ denote all tests that maintain exact $\alpha$-level error control under sequential monitoring for this additive Gaussian noise DGP. Then, the smallest possible sample complexity for our test is given by $\Gamma^{-1}(\tau, f)$, i.e.
>
> $$ \lim_{\alpha\rightarrow 0}\inf_{\xi \in \Xi}\frac{\mathbb{E}\_{P_{N(\tau, \sigma^2)}}[N_f]}{\log(1/\alpha)} \geq \Gamma^{-1}(\tau, f). $$
>
> Our theoretical results show that under asymptotic anytime validity, our tests recover the lower bound corresponding to (i) additive Gaussian noise models with (ii) known conditional variances under (iii) exact error control. Because our methods are applicable to far more general setups than additive Gaussian noise models, our results highlight a practical tradeoff: by relaxing error guarantees, our sample complexity results show that global null testing in general, nonparametric settings is no harder than testing the global null under exact error control constraints for a restricted Gaussian additive noise model with known variances.
>
> To clarify our optimal lower bound statements, we will add a subsection to the end of Section 2 discussing our notion of "optimal" sample complexity with the content above, as well as its implications.
>
>
> **Use of Results from Waudby-Smith et al., 2024**: Our work only leverages Theorem 2.8 of Waudby-Smith et al. (2024) to prove the error control guarantees of Theorem 4.2. For all other theoretical results, including power one and optimal sample complexities, as well as the design of our testing procedure, we **do not leverage results** from Waudby-Smith et al. (2024). We note that Waudby-Smith et al. (2024) focus on asymptotic anytime valid inference for a scalar-valued parameter (e.g. average treatment effect, a scalar mean), with an emphasis on coverage and error control. As a result, their work does not provide any results regarding expected rejection time, unlike the additional theoretical results provided in our work (power one, asymptotic sample complexity bounds). In comparisons to existing works that analyze sample complexities (Bibaut et al., 2024), our work leverages a distinct proof of the sample complexity based on uniform boundedness conditions, rather than multiple moment bounds (see Section 5 of Bibaut et al., 2024 to see the difference in assumptions used for complexity results).

---

> > ### Author Rebuttal · Reviewer_TGWx · 2026-04-04
> >
> > Thank you to the authors for the detailed response. My questions have been adequately addressed and I am maintaining my positive score.

---

### Official Review · Reviewer_GB3B · 2026-03-16

**Soundness:** 3
**Presentation:** 2
**Significance:** 2
**Originality:** 2
**Overall Recommendation:** 4
**Confidence:** 2

**Summary:**

This paper proposes GAAVI, an asymptotic anytime-valid procedure for testing global null hypotheses on conditional mean functions in a sequential setting, allowing continuous monitoring without pre-specifying a sample size.
The authors prove that, under mild assumptions, the method achieves asymptotic anytime-valid type-I error control, attains power one under a positive-covariance condition, and matches the asymptotically optimal sample complexity of an ideal Gaussian benchmark. Empirical results further suggest that GAAVI preserves the nominal error rate while maintaining strong power across diverse data-generating settings.

**Compliance With Llm Reviewing Policy:**

Affirmed.

**Final Justification:**

My questions have been adequately addressed, and I maintain my positive evaluation of the paper.

**Key Questions For Authors:**

1. Could the authors provide more intuition for what “asymptotic anytime-valid” means in practice? Since real applications typically involve finite samples, it would be helpful to see how the readers should interpret this guarantee operationally. Relatedly, how should we think about the practical value of the asymptotic optimal sample complexity result in finite-sample settings, and what does it tell us about the method’s behavior in practice?

2. Could the authors discuss the extent to which the method or theory may extend beyond the i.i.d. setting? In particular, what behavior should one expect under distribution drift over time or under adaptive treatment assignment based on past observations?

**Limitations:**

The paper discusses several directions for future work in the final section, but it would be helpful to more explicitly identify which assumptions are most likely to be violated in practice.

**Strengths And Weaknesses:**

Overall, the paper appears technically sound, and the main claims are supported by substantial theoretical analysis and empirical evidence.
The authors position the work relative to prior literature and show that GAAVI achieves nice performance guarantees under some assumptions.

My main concern is with clarity rather than correctness. As a non-expert reader, I found the narrative somewhat difficult to follow early on, especially in understanding the practical meaning of the global null, as well as the role of the weighted martingale statistic. I am also a bit confused by  the distinction between asymptotic anytime validity and exact anytime validity.
I think the paper would benefit from a more explicit plain-language roadmap in Sections 2–3, along with a table that maps each main theorem to its required assumptions and practical interpretation.

I also think the paper would be strengthened by returning more explicitly to a concrete motivating use case, such as subgroup monitoring in clinical trials or fairness auditing (as the paper mentioned in the introduction, but unfortunately it didn't quite follow through), to help readers connect the mathematical assumptions to realistic applied settings.

While I am not expert enough to fully judge the paper’s originality relative to the specialized literature, the problem setting appears important and potentially useful for practitioners in areas such as medicine and A/B testing.

---

> ### Author Rebuttal · Authors · 2026-03-30
>
> We thank the reviewer for their detailed reading and insightful comments regarding our submission. In particular, we appreciate that the reviewer finds our submission (i) well supported by both theoretical analysis and empirical evidence, (ii) technically sound with performance guarantees, and (iii) well-motivated by the problem setting. Below, we hope to address the questions and comments raised by the reviewer:
>
> **Clarity Edits**:  To ease with exposition, we will make the following changes to the manuscript:
> - To make the definition of global null testing clear, we upgraded the current in-line definition (provided on lines 140-145, RHS) to a "Definition" environment, and provide remarks below that connect this hypothesis with existing works in the literature.
> - To contextualize our definition for asymptotic anytime validity (Definition 2.4), we added a remark below (lines 132-128) that clarifies the difference between asymptotic anytime validity and exact anytime validity. We have added an explicit definition of exact anytime validity to our appendix, with our remark referencing this definition.
> - To contextualize our results with a running example, we will include our real-world experiment JOBS as a motivating example for our method. To incorporate this example, we make the following changes. At the end of Section 2, we introduce the JOBS dataset as our case study, and discuss our assumptions (Assumptions 2.1, 2.2, 2.3) in the context of this study. Below Assumptions 4.3 and 4.5, we discuss each assumption under the context of our JOBS dataset. For example, while Assumption 4.5 (optimal sample complexities) requires the convergence of $\hat\tau_t$ to the true conditional probability of re-employment, Assumption 4.3 (eventual rejection) only needs $\hat\tau_t$ to converge to a limiting function positively correlated with the true reemployment probabilities.
>
> **Key Questions/Limitations**
> - *Intuition for Asymptotic AV*: In practice, asymptotic anytime validity means that we wait beyond a minimum sample size $t_0$ before monitoring for rejection. For sufficiently large $t_0$, asymptotic AV guarantees that error rate is controlled under continuous monitoring beyond $t_0$. The "sufficiently large" criteria can be thought equivalently to inference based on the central limit theorem (e.g. the workhorse behind $\pm 1.96$ standard error confidence intervals, $t$-tests, etc). Note that CLT-based inference only offers fixed-time inference at sufficiently large sample sizes; similarly, our approach enables error protection under continuous monitoring beyond sufficiently large minimum sample sizes. We refer to Figure 6 (Null) in Appendix B to show relatively small $t_0$ still provide error control.
> - *Intuition Behind Asymptotic Sample Complexities*: Because error tolerance $\alpha$ is often set close to zero in real-world settings, the asymptotic sample complexities approximately correspond to $$\mathbb{E}[N_f(\alpha)] \approx \log(1/\alpha) \Gamma^{-1}(\tau, f),$$ and quantify the expected number of samples before rejection. Alternatively, one can view the asymptotic sample complexity as a *rate of information gain*, i.e. the number of additional samples needed for rejection per unit increase of confidence on the $\log(1/\alpha)$ scale. Smaller asymptotic sample complexity simply states that in the long-run, our testing procedure acquires information against the null at the faster rates.
> - *Beyond I.I.D.*: Our method naturally extends beyond i.i.d. settings. In the setting where either $\tau$ or $\pi$ are nonstationary (e.g. a sequence of $(\tau_t)\_{t}$, adaptive sampling policies), one can replace $\pi(X_t, a)$ in $\phi_{t, a}$ with $\pi_t(X_t, a) = P(A=a|X_t, H_{t-1})$ in Eq. (3) to maintain our error control guarantees for the hypothesis $\mathcal{H}: \tau_t = f$ for all $t$. This can be generalized to the hypotheses $\mathcal{H}: \tau_t = f_t  \ \forall t$ for time-dependent $(f_t)_{t \in \mathbb{N}}$ by replacing the term $f$ in Eq. (4) with $f_t$. While power one and sample complexity results follow from similar generalizations to Assumption 4.3 and 4.4, the nonstationary case requires us to estimate a sequence of time-varying functions (e.g. $\tau_t, v_t$). Because estimation heavily depends on the type of nonstationarity, we defer this extension to future work, and believe it is out of scope for our submission.
> - *Violations of Assumptions*: A key benefit of our approach is the relative *laxness* of the assumptions needed relative to existing methods (see Section 1.1). Among our assumptions, Assumption 4.5 may be most likely to be violated due to misspecified model classes, even with flexible ML-based regression methods. For sample complexity results under misspecified limiting nuisance models, we refer to **Nuisance Estimation Considerations** in the rebuttal for Reviewer b2X7. Error control and power one properties hold under much milder conditions, demonstrated by its empirical robustness.

---

> > ### Author Rebuttal · Reviewer_GB3B · 2026-04-03
> >
> > Thank you for the detailed response. My questions have been adequately addressed and I encourage the authors to incorporate this feedback into their revision.

---

### Decision · Program_Chairs · 2026-04-30

**Decision:**

Accept (regular)

**Comment:**

This work studies asymptotically valid sequential testing framework for conditional mean functions that allows for monitoring an experiment (past a minimum sample size) while maintaining type-I error. This is achieved via a function valued confidence sequence. Strong empirical performance is shown for continuous monitoring on both synthetic and real data. Reviewers all felt that this work addresses an important and relevant task, and that the proposed method is very well supported with theoretical guarantees on type-I control, and sample complexity. There is also broad consensus that this work has task novelty providing anytime valid methodology for _conditional_ mean functions in the presence of continuous and high dimensional covariates. The main concerns amongst reviewers were that portions of the paper could be improved for clarity, in particular in the theory portions, that there should be a larger discussion around the required bur in period and its ramifications for finite sample behavior, the sensitivity to error in nuissance estimation, and additional motivating examples. All of these are narrative issues which I encourage the authors to address in order to improve the audience and impact of the work.